# Initialization-Dependent Sample Complexity of Linear Predictors and Neural Networks

**Roey Magen**
Weizmann Institute of Science
`roey.magen@weizmann.ac.il`

**Ohad Shamir**
Weizmann Institute of Science
`ohad.shamir@weizmann.ac.il`

## Abstract

We provide several new results on the sample complexity of vector-valued linear predictors (parameterized by a matrix), and more generally neural networks. Focusing on size-independent bounds, where only the Frobenius norm distance of the parameters from some fixed reference matrix $W_0$ is controlled, we show that the sample complexity behavior can be surprisingly different than what we may expect considering the well-studied setting of scalar-valued linear predictors. This also leads to new sample complexity bounds for feed-forward neural networks, tackling some open questions in the literature, and establishing a new convex linear prediction problem that is provably learnable without uniform convergence.

## 1 Introduction

In this paper, we consider the sample complexity of learning function classes, where each function is a composition of one or more transformations given by

$$\mathbf{x} \ \rightarrow \ f(W\mathbf{x}) \,,$$

where $\mathbf{x}$ is a vector, $W$ is a parameter matrix, and $f$ is some fixed Lipschitz function. A natural example is vanilla feed-forward neural networks, where each such transformation corresponds to a layer with weight matrix $W$ and some activation function $f$. A second natural example are vector-valued linear predictors (e.g., for multi-class problems), where $W$ is the predictor matrix and $f$ corresponds to some loss function. A special case of the above are scalar-valued linear predictors (composed with some scalar loss or nonlinearity $f$), namely $\mathbf{x} \to f(\mathbf{w}^\top \mathbf{x})$, whose sample complexity is extremely well-studied. However, we are interested in the more general case of matrix-valued $W$, which (as we shall see) is far less understood.

Clearly, in order for learning to be possible, we must impose some constraints on the size of the function class. One possibility is to bound the number of parameters (i.e., the dimensions of the matrix W), in which case learnability follows from standard VC-dimension or covering number arguments (see Anthony and Bartlett [1999]). However, an important thread in statistical learning theory is understanding whether bounds on the number of parameters can be replaced by bounds on the magnitude of the weights – say, a bound on some norm of $W$. For example, consider the class of scalar-valued linear predictors of the form

$$\{\mathbf{x} \to \mathbf{w}^\top \mathbf{x} : \mathbf{w}, \mathbf{x} \in \mathbb{R}^d, \|\mathbf{w}\| \le B\}$$

and inputs $\|\mathbf{x}\| \le 1$, where $\| \cdot \|$ is the Euclidean norm. For this class, it is well-known that the sample complexity required to achieve excess error $\epsilon$ (w.r.t. Lipschitz losses) scales as $O(B^2/\epsilon^2)$, independent of the number of parameters $d$ (e.g., Bartlett and Mendelson [2002], Shalev-Shwartz and Ben-David [2014]). Moreover, the same bound holds when we replace $\mathbf{w}^\top \mathbf{x}$ by $f(\mathbf{w}^\top \mathbf{x})$ for some 1-Lipschitz function $f$. Therefore, it is natural to ask whether similar size-independent bounds can be obtained when $W$ is a matrix, as described above. This question is the focus of our paper.

37th Conference on Neural Information Processing Systems (NeurIPS 2023).

When studying the matrix case, there are two complicating factors: The first is that there are many possible generalizations of the Euclidean norm for matrices (namely, matrix norms which reduce to the Euclidean norm in the case of vectors), so it is not obvious which one to study. A second is that rather than constraining the norm of $W$, it is increasingly common in recent years to constrain the distance to some fixed reference matrix $W_0$, capturing the standard practice of non-zero random initialization (see, e.g., Bartlett et al. [2017]). Following a line of recent works in the context of neural networks (e.g., Vardi et al. [2022], Daniely and Granot [2019, 2022]), we will be mainly interested in the case where we bound the *spectral norm* $|| \cdot ||$ of $W_0$, and the distance of $W$ from $W_0$ in the *Frobenius norm* $|| \cdot ||_F$, resulting in function classes of the form

$$\left\{ \mathbf{x} \to f(W\mathbf{x}) : W \in \mathbb{R}^{n \times d}, ||W - W_0||_F \leq B \right\}. \tag{1}$$

for some Lipschitz, possibly non-linear function $f$ and a fixed $W_0$ of bounded spectral norm. This is a natural class to consider, as we know that spectral norm control is necessary (but insufficient) for finite sample complexity guarantees (see, e.g., Golowich et al. [2018]), whereas controlling the (larger) Frobenius norm is sufficient in many cases. Moreover, the Frobenius norm (which is simply the Euclidean norm of all matrix entries) is the natural metric to measure distance from initialization when considering standard gradient methods, and also arises naturally when studying the implicit bias of such methods (see Lyu and Li [2019]). As to $W_0$, we note that in the case of scalar-valued linear predictors (where $W, W_0$ are vectors), the sample complexity is not affected [1] by $W_0$. This is intuitive, since the function class corresponds to a ball of radius $B$ in parameter space, and $W_0$ affects the location of the ball but not its size. A similar weak dependence on $W_0$ is also known to occur in other settings that were studied (e.g., Bartlett et al. [2017]).

In this paper, we provide several new contributions on the size-independent sample complexity of this and related function classes, in several directions.

In the first part of the paper (Section 3), we consider function classes as in Eq. (1), without further assumptions on $f$ besides being Lipschitz, and assuming $\mathbf{x}$ has a bounded Euclidean norm. As mentioned above, this is a very natural class, corresponding (for example) to vector-valued linear predictors with generic Lipschitz losses, or neural networks composed of a single layer and some general Lipschitz activation. In this setting, we make the following contributions:

- In subsection 3.1 we study the case of $W_0 = 0$, and prove that the size-independent sample complexity (up to some accuracy $\epsilon$) is both upper and lower bounded by $2^{\tilde{\Theta}(B^2/\epsilon^2)}$. This is unusual and perhaps surprising, as it implies that this function class does enjoy a finite, size-independent sample complexity bound, but the dependence on the problem parameters $B, \epsilon$ are exponential. This is in very sharp contrast to the scalar-valued case, where the sample complexity is just $O(B^2/\epsilon^2)$ as described earlier. Moreover, and again perhaps unexpectedly, this sample complexity remains the same even if we consider the much larger function class of *all* bounded Lipschitz functions, composed with all norm-bounded linear functions (as opposed to having a single fixed Lipschitz function).

- Building on the result above, we prove a size-independent sample complexity upper bound for deep feed-forward neural networks, which depends only on the Frobenius norm of the first layer, and the product of the spectral norms of the other layers. In particular, it has no dependence whatsoever on the network depth, width or any other matrix norm constraints, unlike previous works in this setting.

- In subsection 3.2, we turn to consider the case of $W_0 \neq 0$, and ask if it is possible to achieve similar size-independent sample complexity guarantees. Perhaps unexpectedly, we show that the answer is no, even for $W_0$ with very small spectral norm. Again, this is in sharp qualitative contrast to the scalar-valued case and other settings in the literature involving a $W_0$ term, where the choice of $W_0$ does not strongly affect the bounds.

- In subsection 3.3, we show that the negative result above yields a new construction of a convex linear prediction problem which is learnable (via stochastic gradient descent), but where uniform convergence provably does not hold. This adds to a well-established line of works in statistical learning theory, studying when uniform convergence is provably unnecessary for distribution-free

---

[1] More precisely, it is an easy exercise to show that the Rademacher complexity of the function class $\{\mathbf{x} \to f(\mathbf{w}^\top \mathbf{x}) : \mathbf{w} \in \mathbb{R}^d, ||\mathbf{w} - \mathbf{w}_0|| \leq B\}$, for some fixed Lipschitz function $f$, can be upper bounded independent of $\mathbf{w}_0$.

learnability (e.g., Shalev-Shwartz et al. [2010], Daniely et al. [2011], Feldman [2016], see also Nagarajan and Kolter [2019], Glasgow et al. [2022] in a somewhat different direction).

In the second part of our paper (Section 4), we turn to a different and more specific choice of the function $f$, considering one-hidden-layer neural networks with activation applied element-wise:

$$\mathbf{x} \longrightarrow \mathbf{u}^\top \sigma(W\mathbf{x}) = \sum_j u_j \sigma(\mathbf{w}_j^\top \mathbf{x}),$$

with weight matrix $W \in R^{n \times d}$, weight vector $\mathbf{u} \in \mathbb{R}^n$, and a fixed (generally non-linear) Lipschitz activation function $\sigma(\cdot)$. As before, We focus on an Euclidean setting, where $\mathbf{x}$ and $\mathbf{u}$ has a bounded Euclidean norm and $||W - W_0||_F$ is bounded, for some initialization $W_0$ with bounded spectral norm. In this part, our sample complexity bounds have polynomial dependencies on the norm bounds and on the target accuracy $\epsilon$. Our contributions here are as follows:

- We prove a fully size-independent Rademacher complexity bound for this function class, under the assumption that the activation $\sigma(\cdot)$ is smooth. In contrast, earlier results that we are aware of were either not size-independent, or assumed $W_0 = 0$. Although we do not know whether the smoothness assumption is necessary, we consider this an interesting example of how smoothness can be utilized in the context of sample complexity bounds.

- With $W_0 = 0$, we show an upper bound on the Rademacher complexity of deep neural networks (more than one layer) that is fully independent of the network width or the input dimension, and for generic element-wise Lipschitz activations. For constant-depth networks, this bound is fully independent of the network size.

These two results answer some of the open questions in Vardi et al. [2022]. We conclude with a discussion of open problems in Section 5. Formal proofs of our results appear in the appendix.

## 1.1 Comparison to Previous Work

**Deep neural networks and the Frobenius norm.** A size-independent uniform convergence guarantee, depending on the product of Frobenius norm of all layers, has been established in Neyshabur et al. [2015] for constant-depth networks, and in Golowich et al. [2018] for arbitrary-depth networks. However, these bounds are specific to element-wise, homogeneous activation functions, whereas we tackle general Lipschitz activations. Bounds based on other norms include Anthony and Bartlett [1999], Bartlett et al. [2017], but are potentially more restrictive than the Frobenius norm, or not independent of the width. All previous bounds of this type (that we are aware of) strongly depend on various norms of all layers, which can be arbitrarily larger than the spectral norm in a size-independent setting (such as the Frobenius norm and the $(1, 2)$-norm), or make strong assumptions on the activation function.

**Rademacher complexity of vector-valued functions.** Maurer [2016] showed a contraction inequality for Rademacher averages that extended to Lipschitz functions with vector-valued domains. As an application, he showed an upper bound on the Rademacher complexity of vector-valued linear predictors. However, his bound depends polynomially on the number of parameters (i.e. on $\sqrt{n}$), where we focus on size-independent bounds, which do not depend on the input's dimension or number of parameters. The sample complexity of vector-valued predictors has also been extensively studied in the context of multiclass classification (see for example Mohri et al. [2018]). However, these results generally depend polynomially on the vector dimension (e.g., number of classes), and are not size-independent.

**Non-zero initialization.** Bartlett et al. [2017] upper bound the sample complexity of neural networks with non-zero initialization, but they used a much stronger assumption than ours: They control the $(1, 2)$-matrix norm (sum of $L_2$-norms of the columns), and the gap between this norm and the Frobenius norm can be arbitrarily large, depending on the matrix size. Daniely and Granot [2022] also recently studied the non-zero initialization case, with element-wise activations and Frobenius norm constraints. However, their results in this case are not size-independent and employ a different proof technique than ours.

**Non-element-wise activations.** Daniely and Granot [2022] provide a fat-shattering lower bound for a general (possibly non-element-wise) Lipchitz activation, which implies that neural networks on $\mathbb{R}^d$

with bounded Frobenius norm and width $n$ can shatter $n$ points with constant margin, assuming that the inputs have norm $\sqrt{d}$ and that $n = O(2^d)$. However, this lower bound does not separate between the input norm bound and the width of the hidden layer. Therefore, their result does not contradict our upper bound (Thm. 2) which implies that it is possible to achieve a size-independent upper bound on the sample complexity, when the input norm is fixed independent of the network width.

## 2 Preliminaries

**Notations.** We use bold-face letters to denote vectors, and let $[m]$ be shorthand for $\{1, 2, \ldots, m\}$. Given a vector $\mathbf{x}$, $x_j$ denotes its $j$-th coordinate. Given a matrix $W$, $\mathbf{w}_j$ is its $j$-th row, and $W_{j,i}$ is its entry in row $j$ and column $i$. Let $0_{n \times d}$ denote the zero matrix in $\mathbb{R}^{n \times d}$, and let $I_{d \times d}$ be the $d \times d$ identity matrix. Given a function $\sigma(\cdot)$ on $\mathbb{R}$, we somewhat abuse notation and let $\sigma(\mathbf{x})$ (for a vector $\mathbf{x}$) or $\sigma(M)$ (for a matrix M) denote applying $\sigma(\cdot)$ element-wise. We use standard big-Oh notation, with $\Theta(\cdot), \Omega(\cdot), O(\cdot)$ hiding constants and $\tilde{\Theta}(\cdot), \tilde{\Omega}(\cdot), \tilde{O}(\cdot)$ hiding constants and factors that are polylogarithmic in the problem parameters.

We let $\|\cdot\|$ denote the operator norm: For vectors, it is the Euclidean norm, and for matrices, the spectral norm (i.e., $\|M\| = \sup_{x:\|\mathbf{x}\|=1} \|M\mathbf{x}\|$). $\|\cdot\|_F$ denotes the Frobenius norm (i.e., $\|M\|_F = \sqrt{\sum_{i,j} M_{i,j}^2}$). It is well-known that the spectral norm of a matrix is equal to its largest singular value, and that the Frobenius norm is equal to $\sqrt{\sum_i \sigma_i^2}$, where $\sigma_1, \sigma_2, \ldots$ are the singular values of the matrix.

When we say that a function $f$ is Lipschitz, we refer to the Euclidean metric unless specified otherwise. We say that $f : \mathbb{R}^d \to \mathbb{R}$ is $\mu$-smooth if $f$ is continuously differentiable and its gradient $\nabla f$ is $\mu$-Lipschitz.

Given a metric space $(\mathcal{X}, d)$ and $\epsilon > 0$, we say that $N \subseteq \mathcal{X}$ is an $\epsilon-$cover for $\mathcal{U} \subseteq \mathcal{X}$ if for every $x \in U$, there exists $x' \in N$ such that $d(x, x') \leq \epsilon$. We say that $P \subseteq \mathcal{X}$ is an $\epsilon-$packing (or $\epsilon-$seperated), if for every $x, x' \in P$ such that $x \neq x'$ we have that $d(x, x') \geq \epsilon$.

**Sample Complexity Measures.** In our results and proofs, we consider several standard complexity measures of a given class of functions, which are well-known to control uniform convergence, and imply upper or lower bounds on the sample complexity:

- Fat-Shattering dimension: It is well-known that the fat-shattering dimension (at scale $\epsilon$) lower bounds the number of samples needed to learn in a distribution-free learning setting, up to accuracy $\epsilon$ (see for example Anthony and Bartlett [2002]). It is formally defined as follows:

  **Definition 1** (Fat-Shattering). *A class of functions $\mathcal{F}$ on an input domain $\mathcal{X}$ shatters $m$ points $\mathbf{x}_1, ..., \mathbf{x}_m \in \mathcal{X}$ with margin $\epsilon$, if there exists a number $s$, such that for all $y \in \{0,1\}^m$ we can find some $f \in \mathcal{F}$ such that*

  $$\forall i \in [m], \ f(\mathbf{x}_i) \leq s - \epsilon \ if \ y_i = 0 \ and \ f(\mathbf{x}_i) \geq s + \epsilon \ if \ y_i = 1$$

  *The fat-shattering dimension of F (at scale $\epsilon$) is the cardinality $m$ of the largest set of points in $\mathcal{X}$ for which the above holds.*

  Thus, by proving the existence of a large set of points shattered by the function class, we get lower bounds on the fat-shattering dimension, which translate to lower bounds on the sample complexity.

- Rademacher Complexity: This measure can be used to obtain upper bounds on the sample complexity: Indeed, the number of inputs $m$ required to make the Rademacher complexity of a function class $\mathcal{F}$ smaller than some $\epsilon$ is generally an upper bound on the number of samples required to learn $\mathcal{F}$ up to accuracy $\epsilon$ (see Shalev-Shwartz and Ben-David [2014]).

  **Definition 2** (Rademacher complexity). *Given a class of functions $\mathcal{F}$ on a domain $\mathcal{X}$, its Rademacher complexity is defined as $R_m(\mathcal{F}) = \sup_{\{\mathbf{x}_i\}_{i=1}^m \subseteq \mathcal{X}} \mathbb{E}_\epsilon \left[ \sup_{f \in \mathcal{F}} \frac{1}{m} \sum_{i=1}^m \epsilon_i f_i(\mathbf{x}_i) \right]$, where $\epsilon = (\epsilon_1, ..., \epsilon_m)$ is uniformly distributed on $\{-1, +1\}^m$.*

- Covering Numbers: This is a central tool in the analysis of the complexity of classes of functions (see, e.g., Anthony and Bartlett [2002]), which we use in our proofs.

  **Definition 3** (Covering Number). *Given any class of functions $\mathcal{F}$ from $\mathcal{X}$ to $\mathcal{Y}$, a metric $d$ over functions from $\mathcal{X}$ to $\mathcal{Y}$, and $\epsilon > 0$, we let the covering number $N(\mathcal{F}, d, \epsilon)$ denote the minimal*

*number $n$ of functions $f_1, f_2, ..., f_n$ from $\mathcal{X}$ to $\mathcal{Y}$, such that for all $f \in \mathcal{F}$, there exists some $f_i$ with $d(f_i, f) \leq \epsilon$. In this case we also say that $\{f_1, f_2, ..., f_n\}$ is an $\epsilon$-cover for $\mathcal{F}$.*

In particular, we will consider covering numbers with respect to the empirical $L_2$ metric defined as $d_m(f, f') = \sqrt{\frac{1}{m} \sum_{i=1}^{m} ||f(\mathbf{x}_i) - f'(\mathbf{x}_i)||^2}$ for some fixed set of inputs $\mathbf{x}_1, \ldots, \mathbf{x}_m$. In addition, if $\{f_1, f_2, ..., f_n\} \subseteq \mathcal{F}$, then we say that this cover is *proper*. It is well known that the distinction between proper and improper covers is minor, in the sense that the proper $\epsilon$-covering number is sandwiched between the improper $\epsilon$-covering number and the improper $\frac{\epsilon}{2}$-covering number (see the appendix for a formal proof):

**Observation 1.** *Let $\mathcal{F}$ be a class of functions. Then the proper $\epsilon$-covering number for $\mathcal{F}$ is at least $N(\mathcal{F}, d, \epsilon)$ and at most $N(\mathcal{F}, d, \frac{\epsilon}{2})$.*

# 3 Linear Predictors and Neural Networks with General Activations

We begin by considering the following simple matrix-parameterized class of functions on $\mathcal{X} = \{\mathbf{x} \in \mathbb{R}^d : ||\mathbf{x}|| \leq 1\}$:

$$\mathcal{F}_{B,n,d}^{f,W_0} := \left\{ \mathbf{x} \to f(W\mathbf{x}) : W \in \mathbb{R}^{n \times d}, ||W - W_0||_F \leq B \right\} ,$$

where $f$ is assumed to be some fixed $L$-Lipschitz function, and $W_0$ is a fixed matrix in $\mathbb{R}^{n \times d}$ with a bounded spectral norm. As discussed in the introduction, this can be interpreted as a class of vector-valued linear predictors composed with some Lipschitz loss function, or alternatively as a generic model of one-hidden-layer neural networks with a generic Lipschitz activation function. Moreover, $W_0$ denotes an initialization/reference point which may or may not be 0.

In this section, we study the sample complexity of this class (via its fat-shattering dimension for lower bounds, and Rademacher complexity for upper bounds). Our focus is on size-independent bounds, which do not depend on the input dimension $d$ or the matrix size/network width $n$. Nevertheless, to understand the effect of these parameters, we explicitly state the conditions on $d, n$ necessary for the bounds to hold.

**Remark 1.** *Enforcing $f$ to be Lipschitz and the domain $\mathcal{X}$ to be bounded is known to be necessary for meaningful size-independent bounds, even in the case of scalar-valued linear predictors $\mathbf{x} \mapsto f(\mathbf{w}^\top \mathbf{x})$ (e.g., Shalev-Shwartz and Ben-David [2014]). For simplicity, we mostly focus on the case of $\mathcal{X}$ being the Euclidean unit ball, but this is without much loss of generality: For example, if we consider the domain $\{\mathbf{x} \in \mathbb{R}^d : ||\mathbf{x}|| \leq b_x\}$ in Euclidean space for some $b_x \geq 0$, we can embed $b_x$ into the weight constraints, and analyze instead the class $\mathcal{F}_{b_x B,n,d}^{f, b_x W_0}$ over the Euclidean unit ball $\{\mathbf{x} \in \mathbb{R}^d : ||\mathbf{x}|| \leq 1\}$.*

## 3.1 Size-Independent Sample Complexity Bounds with $W_0 = 0$

First, we study the case of initialization at zero (i.e. $W_0 = 0_{n \times d}$). Our first lower bound shows that the size-independent fat-shattering dimension of $\mathcal{F}_{B,n,d}^{f,W_0}$ (at scale $\epsilon$) is at least exponential in $B^2/\epsilon^2$:

**Theorem 1.** *For any $B, L \geq 1$ and $\epsilon \in (0, 1]$ s.t. $\frac{L^2 B^2}{128\epsilon^2} \geq 20$, there exists large enough $d = \Theta(L^2 B^2/\epsilon^2), n = \exp(\Theta(L^2 B^2/\epsilon^2))$ and an $L$-Lipschitz function $f : \mathbb{R}^n \to \mathbb{R}$ for which $\mathcal{F}_{B,n,d}^{f,W_0=0}$ can shatter*

$$\exp(cL^2 B^2/\epsilon^2)$$

*points from $\{\mathbf{x} \in \mathbb{R}^d : ||\mathbf{x}|| \leq 1\}$ with margin $\epsilon$, where $c > 0$ is a universal constant.*

The proof is directly based on the proof technique of Theorem 3 in Daniely and Granot [2022], and differs from them mainly in that we focus on the dependence on $B, \epsilon$ (whereas they considered the dependence on $n, d$). The main idea is to use the probabilistic method to show the existence of $\mathbf{x}_1, ..., \mathbf{x}_m \in \mathbb{R}^d$ and $W_1, ..., W_{2^m} \in \mathbb{R}^{n \times d}$ for $m = \Theta(B^2/\epsilon^2)$, with the property that every two different vectors from $\{W_y \mathbf{x}_i : i \in m, y \in [2^m]\}$ are far enough from each other. We then construct an $L$-Lipschitz function $f$ which assigns arbitrary outputs to all these points, resulting in a shattering as we range over $W_1, \ldots W_{2^m}$.

We now turn to our more novel contribution, which shows that the bound above is nearly tight, in the sense that we can upper bound the Rademacher complexity of the function class by a similar quantity. In fact, and perhaps surprisingly, a much stronger statement holds: A similar quantity upper bounds the complexity of the much larger class of *all $L$-Lipschitz function $f$* on $\mathbb{R}^n$, composed with all norm-bounded linear functions from $\mathbb{R}^d$ to $\mathbb{R}^n$:

**Theorem 2.** *For any $L, B \geq 1$ and $\epsilon \in (0, 1]$ s.t. $\frac{LB}{\epsilon} \geq 1$, let $\Psi_{L,a,n}$ be the class of all $L$-Lipschitz functions from $\{\mathbf{x} \in \mathbb{R}^n : ||\mathbf{x}|| \leq B\}$ to $\mathbb{R}$, which equal some fixed $a \in \mathbb{R}$ at $\mathbf{0}$. Let $\mathcal{W}_{B,n}$ be the class of linear functions from $\mathbb{R}^d$ to $\mathbb{R}^n$ over the domain $\{\mathbf{x} \in \mathbb{R}^d : ||\mathbf{x}|| \leq 1\}$ with Frobenius norm at most $B$, namely*

$$\mathcal{W}_{B,n} := \{\mathbf{x} \to W\mathbf{x} : W \in \mathbb{R}^{n \times d}, ||W||_F \leq B\}.$$

*Then the Rademacher complexity of $\Psi_{a,L,n} \circ \mathcal{W}_{B,n} := \{\psi \circ g : \psi \in \Psi_{L,a,n}, g \in \mathcal{W}_{B,n}\}$ on $m$ inputs from $\{\mathbf{x} \in \mathbb{R}^d : ||\mathbf{x}|| \leq 1\}$ is at most $\epsilon$, if $m \geq \left(\frac{LB}{\epsilon}\right)^{\frac{cL^2B^2}{\epsilon^2}}$ for some universal constant $c > 0$.*

Since $\mathcal{F}_{B,n,d}^{f, W_0 = 0} \subseteq \Psi_{L,a,n} \circ \mathcal{W}_{B,n}$ for any fixed $f$, the Rademacher complexity of the latter upper bounds the Rademacher complexity of the former. Thus, we get the following corollary:

**Corollary 1.** *Let $f : \mathbb{R}^n \to \mathbb{R}$ be a fixed $L$-Lipschitz function. Then the Rademacher complexity of $\mathcal{F}_{B,n,d}^{f, W_0 = 0}$ on $m$ inputs from $\{\mathbf{x} \in \mathbb{R}^d : ||\mathbf{x}|| \leq 1\}$ is at most $\epsilon$, if $m \geq \left(\frac{LB}{\epsilon}\right)^{\frac{cL^2B^2}{\epsilon^2}}$ for some universal constant $c > 0$.*

Comparing the corollary and Theorem 1, we see that the sample complexity of Lipschitz functions composed with matrix linear ones is $\exp\left(\tilde{\Theta}(L^2B^2/\epsilon^2)\right)$ (regardless of whether the Lipschitz function is fixed in advance or not). On the one hand, it implies that the complexity of this class is very large (exponential) as a function of $L, B$ and $\epsilon$. On the other hand, it implies that for any fixed $L, B, \epsilon$, it is finite completely independent of the number of parameters. The exponential dependence on the problem parameters is rather unusual, and in sharp contrast to the case of scalar-valued predictors (that is, functions of the form $\mathbf{x} \mapsto f(\mathbf{w}^\top \mathbf{x})$ where $||\mathbf{w} - \mathbf{w}_0|| \leq B$ and $f$ is $L$-Lipschitz ), for which the sample complexity is just $O(L^2B^2/\epsilon^2)$.

The key ideas in the proof of Theorem 2 can be roughly sketched as follows: First, we show that due to the Frobenius norm constraints, every function $\mathbf{x} \mapsto f(W\mathbf{x})$ in our class can be approximated (up to some $\epsilon$) by a function of the form $\mathbf{x} \mapsto f(\tilde{W}_\epsilon \mathbf{x})$, where the rank of $\tilde{W}_\epsilon$ is at most $B^2/\epsilon^2$. In other words, this approximating function can be written as $f(UV\mathbf{x})$, where $V$ maps to $\mathbb{R}^{B^2/\epsilon^2}$. Equivalently, this can be written as $g(V\mathbf{x})$, where $g(z) = f(Uz)$ over $\mathbb{R}^{B^2/\epsilon^2}$. This reduces the problem to bounding the complexity of the function class which is the composition of all linear functions to $\mathbb{R}^{B^2/\epsilon^2}$, and all Lipschitz functions over $\mathbb{R}^{B^2/\epsilon^2}$, which we perform through covering numbers and the following technical result:

**Lemma 1.** *Let $\mathcal{F}$ be a class of functions from Euclidean space to $\{\mathbf{x} \in \mathbb{R}^r : ||\mathbf{x}|| \leq B\}$. Let $\Psi_{L,a}$ be the class of all $L$-Lipschitz functions from $\{\mathbf{x} \in \mathbb{R}^r : ||\mathbf{x}|| \leq B\}$ to $\mathbb{R}$, which equal some fixed $a \in \mathbb{R}$ at $\mathbf{0}$. Letting $\Psi_{L,a} \circ \mathcal{F} := \{\psi \circ f : \psi \in \Psi_{L,a}, f \in \mathcal{F}\}$, its covering number satisfies*

$$\log N(\Psi_{L,a} \circ \mathcal{F}, d_m, \epsilon) \leq \left(1 + \frac{8BL}{\epsilon}\right)^r \cdot \log \frac{8BL}{\epsilon} + \log N(\mathcal{F}, d_m, \frac{\epsilon}{4L}).$$

The proof for this Lemma is an extension of Theorem 4 from Golowich et al. 2018, which considered the case $r = 1$. We emphasize that the exponential dependence on $r$ arises from the covering of the class $\Psi_{L,a}$, which is achieved by covering its domain $\{x \in \mathbb{R}^r : ||x|| \leq B\}$ by a set of points $N_x$ of size $(1 + BL/\epsilon)^r$ and covering its range $[a - LB, a + LB]$ by a set $N_y$ of size $2LB/\epsilon$. As we will show, this implies that the set of all functions from $N_x$ to $N_y$ is a cover for $\Psi_{L,a}$.

**Application to Deep Neural Networks.** Theorem 2 which we established above can also be used to study other types of predictor classes. In what follows, we show an application to deep neural networks, establishing a size/dimension-independent sample complexity bound that depends *only* on the Frobenius norm of the first layer, and the spectral norms of the other layers (albeit exponentially). This is surprising, since all previous bounds of this type we are aware of strongly depend on various norms of all layers, which can be arbitrarily larger than the spectral norm in a size-independent setting

(such as the Frobenius norm), or made strong assumptions on the activation function (e.g., Neyshabur et al. [2015, 2017], Bartlett et al. [2017], Golowich et al. [2018], Du and Lee [2018], Daniely and Granot [2019], Vardi et al. [2022]).

Formally, we consider scalar-valued depth-$k$ "neural networks" of the form

$$\mathcal{F}_k := \{x \to \mathbf{w}_k f_{k-1}(W_{k-1} f_{k-2}(...f_1(W_1 \mathbf{x}))) \ : \ ||\mathbf{w}_k|| \leq S_k \ , \ \forall j \ ||W_j|| \leq S_j \ , \ ||W_1||_F \leq B\}$$

where each $W_j$ is a parameter matrix of some arbitrary dimensions, $\mathbf{w}_k$ is a vector and each $f_j$ is some fixed 1-Lipschitz[2] function satisfying $f_j(\mathbf{0}) = 0$. This is a rather relaxed definition for neural networks, as we do not assume anything about the activation functions $f_j$, except that it is Lipschitz. To analyze this function class, we consider $\mathcal{F}_k$ as a subset of the class

$$\left\{ x \to f(W\mathbf{x}) : \|W\|_F \leq B \ , \ f : \mathbb{R}^n \to \mathbb{R} \text{ is } L\text{-Lipschitz} \right\} ,$$

where $L = \prod_{j=2}^k S_j$ (as this clearly upper bounds the Lipschitz constant of $z \mapsto \mathbf{w}_k f_{k-1}(W_{k-1} f_{k-2}(\ldots f_1(z) \ldots))$. By applying Theorem 2 (with the same conditions) we have

**Corollary 2.** *For any $B, L \geq 1$ and $\epsilon \in (0, 1]$ s.t. $\frac{B}{\epsilon} \geq 1$, we have that the Rademacher complexity of $\mathcal{F}_k$ on $m$ inputs from $\{\mathbf{x} \in \mathbb{R}^d : \|\mathbf{x}\| \leq 1\}$ is at most $\epsilon$, if*

$$m \geq \left( \frac{LB}{\epsilon} \right)^{\frac{cL^2 B^2}{\epsilon^2}}$$

*where $L := \prod_{j=2}^k S_j$ and $c > 0$ is some universal constant.*

Of course, the bound has a bad dependence on the norm of the weights, the Lipschitz parameter and $\epsilon$. On the other hand, it is finite for any fixed choice of these parameters, fully independent of the network depth, width, nor on any matrix norm other than the spectral norms, and the Frobenius norm of the first layer only. We note that in the size-independent setting, controlling the product of the spectral norms is both necessary and not sufficient for finite sample complexity bounds (see discussion in Vardi et al. [2022]). The bound above is achieved only by controlling in addition the Frobenius norm of the first layer.

## 3.2 No Finite Sample Complexity with $W_0 \neq 0$

In subsection 3.1, we showed size-independent sample complexity bounds when the initialization/reference matrix $W_0$ is zero. Therefore, it is natural to ask if it is possible to achieve similar size-independent bounds with non-zero $W_0$. In this subsection we show that perhaps surprisingly, the answer is negative: Even for very small non-zero $W_0$, it is impossible to control the sample complexity of $\mathcal{F}_{B,n,d}^{f,W_0}$ independent of the size/dimension parameters $d, n$. Formally, we have the following theorem:

**Theorem 3.** *For any $m \in \mathbb{N}$ and $\epsilon \in (0, \frac{1}{4}]$, there exists $d = m + 1, n = 2m$, $W_0 \in \mathbb{R}^{n \times d}$ with $\|W_0\| = 2\sqrt{2} \cdot \epsilon$ and a function $f : \mathbb{R}^n \to \mathbb{R}$ which is 1-Lipschitz, for which $\mathcal{F}_{B=1,n,d}^{f,W_0}$ can shatter $m$ points from $\{\mathbf{x} \in \mathbb{R}^d : \|\mathbf{x}\| \leq 1\}$ with margin $\epsilon$.*

The theorem strengthens the lower bound of Daniely and Granot [2022] and the previous subsection, which only considered the $W_0 = 0$ case. We emphasize that the result holds already when $\|W_0\|$ is very small (equal to $2\sqrt{2} \cdot \epsilon$). Moreover, the proof technique can be used to show a similar result even if we allow for functions Lipschitz w.r.t. the infinity norm (and not just the Euclidean norm as we have done so far), at the cost of a higher required value of $n$. This is of interest, since non-element-wise activations used in practice (such as variants of the max function) are Lipschitz with respect to that norm, and some previous work utilized such stronger Lipschitz constraints to obtain sample complexity guarantees (e.g., Daniely and Granot [2019]).

Interestingly, the proof of the theorem is simpler than the $W_0 = 0$ case, and involves a direct non-probabilistic construction. It can be intuitively described as follows: We choose a fixed set of

---

[2]This is without loss of generality, since if $f_j$ is $L_j$-Lipschitz, we can rescale it by $1/L_j$ and multiply $S_{j+1}$ by $L_j$.

vectors $\mathbf{x}_1, \ldots, \mathbf{x}_m$ and a matrix $W_0$ (essentially the identity matrix with some padding) so that $W_0 \mathbf{x}_i$ encodes the index $i$. For any choice of target values $\mathbf{y} \in \{\pm \epsilon\}^m$, we define a matrix $W'_{\mathbf{y}}$ (which is all zeros except the values of a half column that are located in a strategic location), so that $W'_{\mathbf{y}} \mathbf{x}_i$ encodes the entire vector $\mathbf{y}$. Letting $W_{\mathbf{y}} = W'_{\mathbf{y}} + W_0$, we get a matrix of bounded distance to $W_0$, so that $W_{\mathbf{y}} \mathbf{x}_i$ encodes both $i$ and $\mathbf{y}$. Thus, we just need $f$ to be the fixed function that given an encoding for $\mathbf{y}$ and $i$, returns $y_i$, hence $\mathbf{x} \mapsto f(W_{\mathbf{y}} \mathbf{x})$ shatters the set of points.

## 3.3  Vector-valued Linear Predictors are Learnable without Uniform Convergence

The class $\mathcal{F}_{B,n,d}^{f,W_0}$, which we considered in the previous subsection, is closely related to the natural class of matrix-valued linear predictors $(\mathbf{x} \mapsto W\mathbf{x})$ with bounded Frobenius distance from initialization, composed with some Lipschitz loss function $\ell$. We can formally define this class as

$$\mathcal{G}_{B,n,d}^{\ell,W_0} := \left\{ (\mathbf{x}, y) \to \ell(W\mathbf{x}; y) \ : \ W \in \mathbb{R}^{n \times d}, \|W - W_0\|_F \le B \right\} \ .$$

For example, standard multiclass linear predictors fall into this form. Note that when $y$ is fixed, this is nothing more than the class $\mathcal{F}_{B,n,d}^{\ell_y,W_0}$ where $\ell_y(z) = \ell(z; y)$. The question of learnability here boils down to the question of whether, given an i.i.d. sample $\{\mathbf{x}_i, y_i\}_{i=1}^m$ from an unknown distribution, we can approximately minimize $\mathbb{E}_{(\mathbf{x},y)}[\ell(W\mathbf{x}, y)]$ arbitrarily well over all $W : \|W - W_0\| \le B$, provided that $m$ is large enough.

For multiclass linear predictors, it is natural to consider the case where the loss $\ell$ is also convex in its first argument. In this case, we can easily establish that the class $\mathcal{G}_{B,n,d}^{\ell,W_0}$ is learnable with respect to inputs of bounded Euclidean norm, regardless of the size/dimension parameters $n, d$. This is because for each $(\mathbf{x}, y)$, the function $W \mapsto \ell(W\mathbf{x}; y)$ is convex and Lipschitz in $W$, and the domain $\{W : \|W - W_0\|_F \le B\}$ is bounded. Therefore, we can approximately minimize $\mathbb{E}_{(\mathbf{x},y)}[\ell(W\mathbf{x}, y)]$ by applying stochastic gradient descent (SGD) over the sequence of examples $\{\mathbf{x}_i, y_i\}_{i=1}^m$. This is a consequence of well-known results (see for example Shalev-Shwartz and Ben-David [2014]), and is formalized as follows:

**Theorem 4.** *Suppose that for any $y$, the function $\ell(., y)$ is convex and $L$-Lipschitz. For any $B > 0$ and fixed matrix $W_0$, there exists a randomized algorithm (namely stochastic gradient descent) with the following property: For any distribution over $(\mathbf{x}, y)$ such that $\|\mathbf{x}\| \le 1$ with probability 1, given an i.i.d. sample $\{(\mathbf{x}_i, y_i)\}_{i=1}^m$, the algorithm returns a matrix $\hat{W}$ such that $\|\hat{W} - W_0\|_F \le B$ and*

$$\mathbb{E}_{\hat{W}} \left[ \mathbb{E}_{(\mathbf{x},y)}[\ell(\hat{W}\mathbf{x}; y)] - \min_{W : \|W - W_0\|_F \le B} \mathbb{E}_{(\mathbf{x},y)}[\ell(W\mathbf{x}; y)] \right] \ \le \ \frac{BL}{\sqrt{m}}.$$

*Thus, the number of samples $m$ required to make the above less than $\epsilon$ is at most $\frac{B^2 L^2}{\epsilon^2}$.*

Perhaps unexpectedly, we now turn to show that this positive learnability result is *not* due to uniform convergence: Namely, we can learn this class, but not because the empirical average and expected value of $\ell(W\mathbf{x}; y)$ are close uniformly over all $W : \|W - W_0\| \le B$. Indeed, that would have required that a uniform convergence measure such as the fat-shattering dimension of our class would be bounded. However, this turns out to be false: The class $\mathcal{G}_{B,n,d}^{\ell,W_0}$ can shatter arbitrarily many points of norm $\le 1$, and at any scale $\epsilon \le 1$, for some small $W_0$ and provided that $n, d$ are large enough[3]. In the previous section, we already showed such a result for the class $\mathcal{F}_{B,n,d}^{f,W_0}$, which equals $\mathcal{G}_{B,n,d}^{\ell,W_0}$ when $y$ is fixed and $f(W\mathbf{x}) = \ell(W\mathbf{x}; y)$. Thus, it is enough to prove that the same impossibility result (Theorem 3) holds even if $f$ is a convex function. This is indeed true using a slightly different construction:

**Theorem 5.** *For any $m \in \mathbb{N}$ and $\epsilon \in (0, \frac{1}{4}]$, there exists large enough $d = \Theta(m), n = \Theta(\exp(m))$, $W_0 \in \mathbb{R}^{n \times d}$ with $\|W_0\| = 4\sqrt{2} \cdot \epsilon$ and a **convex** function $f : \mathbb{R}^n \to \mathbb{R}$ which is 1-Lipschitz with*

---

[3]This precludes uniform convergence, since it implies that for any $m$, we can find a set of $2m$ points $\{\mathbf{x}_i, y_i\}_{i=1}^{2m}$, such that if we sample $m$ points with replacement from a uniform distribution over this set, then there is always some $W$ in the class so that the average value of $\ell(W\mathbf{x}; y)$ over the sample and in expectation differs by a constant independent of $m$. The fact that the fat-shattering dimension is unbounded does not contradict learnability here, since our goal is to minimize the expectation of $\ell(W\mathbf{x}; y)$, rather than view it as predicted values which are then composed with some other loss.

*respect to the infinity norm (and hence also with respect to the Euclidean norm), for which $\mathcal{F}^{f,W_0}_{B=1,n,d}$ can shatter $m$ points from $\{\mathbf{x} \in \mathbb{R}^d : ||\mathbf{x}|| \leq 1\}$ with margin $\epsilon$.*

Overall, we see that the problem of learning vector-valued linear predictors, composed with some convex Lipschitz loss (as defined above), is possible using a certain algorithm, but without having uniform convergence. This connects to a line of work establishing learning problems which are provably learnable without uniform convergence (such as Shalev-Shwartz et al. [2010], Feldman [2016]). However, whether these papers considered synthetic constructions, we consider an arguably more natural class of linear predictors of bounded Frobenius norm, composed with a convex Lipschitz loss over stochastic inputs of bounded Euclidean norm. In any case, this provides another example for when learnability can be achieved without uniform convergence.

## 4    Neural Networks with Element-Wise Lipschitz Activation

In section 3 we studied the complexity of functions of the form $\mathbf{x} \mapsto f(W\mathbf{x})$ (or possibly deeper neural networks) where nothing is assumed about $f$ besides Lipschitz continuity. In this section, we consider more specifically functions which are applied element-wise, as is common in the neural networks literature. Specifically, we will consider the following hypothesis class of scalar-valued, one-hidden-layer neural networks of width $n$ on inputs in $\{\mathbf{x} \in \mathbb{R}^d : ||\mathbf{x}|| \leq b_x\}$, where $\sigma(\cdot)$ is a Lipschitz function on $\mathbb{R}$ applied element-wise, and where we only bound the norms as follows:

$$\mathcal{F}^{\sigma,W_0}_{b,B,n,d} := \left\{ \mathbf{x} \to \mathbf{u}^\top \sigma(W\mathbf{x}) : \mathbf{u} \in \mathbb{R}^n, W \in \mathbb{R}^{n \times d}, ||\mathbf{u}|| \leq b, ||W - W_0||_F \leq B \right\} ,$$

where $\mathbf{u}^\top \sigma(W\mathbf{x}) = \sum_j u_j \sigma(\mathbf{w}_j^\top \mathbf{x})$. We note that we could have also considered a more general version, where $\mathbf{u}$ is also initialization-dependent: Namely, where the constraint $||\mathbf{u}|| \leq b$ is replaced by $||\mathbf{u} - \mathbf{u}_0|| \leq b$ for some fixed $\mathbf{u}_0$. However, this extension is rather trivial, since for vectors $\mathbf{u}$ there is no distinction between the Frobenius and spectral norms. Thus, to consider $\mathbf{u}$ in some ball of radius $b$ around some $\mathbf{u}_0$, we might as well consider the function class displayed above with the looser constraint $||\mathbf{u}|| \leq b + ||\mathbf{u}_0||$. This does not lose much tightness, since such a dependence on $||\mathbf{u}_0||$ is also necessary (see remark 2 below).

The sample complexity of $\mathcal{F}^{\sigma,W_0}_{b,B,n,d}$ was first studied in the case of $W_0 = 0$, with works such as Neyshabur et al. [2015, 2017], Du and Lee [2018], Golowich et al. [2018], Daniely and Granot [2019] proving bounds for specific families of the activation $\sigma(\cdot)$ (e.g., homogeneous or quadratic). For general Lipschitz $\sigma(\cdot)$ and $W_0 = 0$, Vardi et al. [2022] proved that the Rademacher complexity of $\mathcal{F}^{\sigma,W_0=0}_{b,B,n,d}$ for any $L$-Lipschitz $\sigma(\cdot)$ is at most $\epsilon$, if the number of samples is $\tilde{O}\left(\left(\frac{bBb_xL}{\epsilon}\right)^2\right)$. They left the case of $W_0 \neq 0$ as an open question. In a recent preprint, Daniely and Granot [2022] used an innovative technique to prove a bound in this case, but not a fully size-independent one (there remains a logarithmic dependence on the network width $n$ and the input dimension $d$). In what follows, we prove a bound which handles the $W_0 \neq 0$ case and is fully size-independent, under the assumption that $\sigma(\cdot)$ is smooth. The proof (which is somewhat involved) involves techniques different from both previous papers, and may be of independent interest.

**Theorem 6.** *Suppose $\sigma(\cdot)$ (as function on $\mathbb{R}$) is $L$-Lipschitz , $\mu$-smooth (i.e, $\sigma'(\cdot)$ is $\mu$-Lipchitz) and $\sigma(0) = 0$. Then for any $b, B, n, D, \epsilon > 0$ such that $Bb_x \geq 2$, and any $W_0$ such that $||W_0|| \leq B_0$, the Rademacher complexity of $\mathcal{F}^{\sigma,W_0}_{b,B,n,d}$ on $m$ inputs from $\{\mathbf{x} \in \mathbb{R}^d : ||\mathbf{x}|| \leq b_x\}$ is at most $\epsilon$, if*

$$m \geq \frac{1}{\epsilon^2} \cdot \tilde{O}\left(\left(1 + bb_x(LB_0 + (\mu + L)B(1 + B_0 b_x))\right)^2\right).$$

Thus, we get a sample complexity bounds that depend on the norm parameters $b, b_x, B_0$, the Lipschitz parameter $L$, and the smoothness parameter $\mu$, but is fully independent of the size parameters $n, d$. Note that for simplicity, the bound as written above hides some factors logarithmic in $m, B, L, b_x$ – see the proof in the appendix for the precise expression.

We note that if $W_0 = 0$, we can take $B_0 = 0$ in the bound above, in which case the sample complexity scales as $\tilde{O}(((\mu + L)bb_xB)^2/\epsilon^2)$. This is is the same as in Vardi et al. [2022] (see above) up to the dependence on the smoothness parameter $\mu$.

**Remark 2.** *The upper bound on Theorem 6 depends quadratically on the spectral norm of $W_0$ (i.e., $B_0$). This dependence is necessary in general. Indeed, even by taking the activation function $\sigma(\cdot)$ to*

*be the identity, $B = 0$ and $\underline{b} = 1$ we get that our function class contains the class of scalar-valued linear predictors $\{\mathbf{x} \rightarrow \mathbf{v}^\top \mathbf{x} : \mathbf{x}, \mathbf{v} \in \mathbb{R}^d, ||\mathbf{v}|| \leq B_0\}$. For this class, it is well known that the number of samples should be $\Theta(\frac{B_0^2}{\epsilon^2})$, to ensure that the Rademacher complexity of that class is at most $\epsilon$.*

### 4.1 Bounds for Deep Networks with Lipschitz Activations

As a final contribution, we consider the case of possibly deep neural networks, when $W_0 = 0$ and the activations are Lipschitz and element-wise. Specifically, given the domain $\{\mathbf{x} \in \mathbb{R}^d : ||\mathbf{x}|| \leq b_x\}$ in Euclidean space, we consider the class of scalar-valued neural networks of the form

$$\mathbf{x} \rightarrow \mathbf{w}_k^\top \sigma_{k-1}(W_{k-1}\sigma_{k-2}(...\sigma_1((W_1\mathbf{x}))))$$

where $\mathbf{w}_k$ is a vector (i.e. the output of the function is in $\mathbb{R}$) with $||\mathbf{w}_k|| \leq b$, each $W_j$ is a parameter matrix s.t. $||W_j||_F \leq B_j$, $||W_j|| \leq S_j$ and each $\sigma_j(\cdot)$ (as a function on $\mathbb{R}$) is an $L$-Lipschitz function applied element-wise, satisfying $\sigma_j(0) = 0$. Let $\mathcal{F}_{k,\{S_j\},\{B_j\}}^{\{\sigma_j\}}$ be the class of neural networks as above. Vardi et al. [2022] proved a sample complexity guarantee for $k = 2$ (one-hidden-layer neural networks), and left the case of higher depths as an open problem. The theorem below addresses this problem, using a combination of their technique and a "peeling" argument to reduce the complexity bound of networks of depth $k$ to those of depth $(k-1)$. The resulting bound is fully independent of the network width (although strongly depends on the network depth), and is the first of this type (to the best of our knowledge) that handles general Lipschitz activations under Frobenius norm constraints.

**Theorem 7.** *For any $\epsilon, b > 0, \{B_j\}_{j=1}^{k-1}, \{S_j\}_{j=1}^{k-1}, L$ with $S_1, ..., S_{k-1}, L \geq 1$, the Rademacher complexity of $\mathcal{F}_{k,\{S_j\},\{B_j\}}^{\{\sigma_j\}}$ on $m$ inputs from $\{\mathbf{x} \in \mathbb{R}^d : ||\mathbf{x}|| \leq b_x\}$ is at most $\epsilon$, if*

$$m \geq \frac{c \left(kL^{k-1}bR_{k-2}\log^{\frac{3(k-1)}{2}}(m) \cdot \prod_{i=1}^{k-1} B_i\right)^2}{\epsilon^2},$$

*where $R_{k-2} = b_x L^{k-2} \prod_{i=1}^{k-2} S_i$, $R_0 = b_x$ and $c > 0$ is a universal constant.*

We note that for $k = 2$, this reduces to the bound of Vardi et al. [2022] for one-hidden-layer neural networks.

## 5 Discussion and Open Problems

In this paper, we provided several new results on the sample complexity of vector-valued linear predictors and feed-forward neural networks, focusing on size-independent bounds and constraining the distance from some reference matrix. The paper leaves open quite a few avenues for future research. For example, in Section 3, we studied the sample complexity of $\mathcal{F}_{B,n,d}^{f,W_0}$ when $n, d$ are unrestricted. Can we get a full picture of the sample complexity when $n, d$ are also controlled? Even more specifically, can the lower bounds in the section be obtained for any smaller values of $d, n$? As to the results in Section 4, is our Rademacher complexity bound for $\mathcal{F}_{b,B,n,d}^{\sigma,W_0}$ (one-hidden-layer networks and smooth activations) the tightest possible, or can it be improved? Also, can we generalize the result to arbitrary Lipschitz activations? In addition, what is the sample complexity of such networks when $n, d$ are also controlled? In a different direction, it would be very interesting to extend the results of this section to deeper networks and non-zero $W_0$.

## Acknowledgements

This research is supported in part by European Research Council (ERC) grant 754705, by the Israeli Council for Higher Education (CHE) via the Weizmann Data Science Research Center, and by a research grants from the Estate of Louise Yasgour.

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

# A   Proofs

## A.1   Proof of Observation 1

The first part follows immediately from the definition. For the second part, let $k := N(\mathcal{F}, d, \frac{\epsilon}{2})$ and $N = \{g_1, ..., g_k\}$ be an $\frac{\epsilon}{2}$-cover for $\mathcal{F}$. For every $g_i \in N$, let $f_i \in \mathcal{F}$ be some function such that $d(f_i, g_i) \leq \frac{\epsilon}{2}$: there must exist $f_i$ with this property, otherwise we can remove $g_i$ from $N$ and still get an $\frac{\epsilon}{2}$-cover for $\mathcal{F}$, contradicting the minimality property of the covering number. For every $f \in \mathcal{F}$, let $g_i \in N$ be the function with $d(g_i, f) \leq \frac{\epsilon}{2}$, then by the triangle inequality $d(f_i, f) \leq d(f_i, g_i) + d(g_i, f) \leq \epsilon$. Therefore $\{f_1, ..., f_k\}$ is a proper $\epsilon$-cover for $\mathcal{F}$.

## A.2   Proof of Theorem 1

We use the following lemmas from Daniely and Granot [2022]:

**Lemma 2.** *For $d \geq 20$, there exists a set of vectors $\mathbf{x}_1, ..., \mathbf{x}_m \in \mathbb{R}^d$ and a set of matrices $W_1, ..., W_{2^m} \in \mathbb{R}^{n \times d}$ with $n = \Theta(\exp(d))$ that have the following properties:*

1. *$||\mathbf{x}_i|| = 1$, for each $i \in [m]$*

2. *$m = \frac{n}{10}$*

3. *$||W_s||_F^2 \leq 2d$, for each $s \in [2^m]$*

4. *$||W_s \mathbf{x}_i - W_t \mathbf{x}_j|| \geq \frac{1}{4}$, for each $i, j \in [m]$ and $s, t \in [2^m]$ s.t. $(s,i) \neq (t,j)$*

**Lemma 3.** *Let $\mathbf{x}_1, \ldots, \mathbf{x}_m$ be a finite set of different points in some metric space $(\mathcal{X}, d)$, such that for each $i \neq j \in [m]$, $d(\mathbf{x}_i, \mathbf{x}_j) \geq \alpha$. Let further be $p_1, \ldots, p_m \in \mathbb{R}$ any set of points. Then there exists an $L$-Lipschitz function, $f : \mathcal{X} \to \mathbb{R}$, where*

$$L = \frac{2}{\alpha} \min_{C \geq 0} \max_{i \in [m]} |p_i - C|,$$

*such that for each $i \in [m]$, $f(\mathbf{x}_i) = p_i$.*

*Proof of Theorem 1.* Let $B \geq 1$, $\epsilon \leq 1$, and $d$ that will be chosen later. Let $\mathbf{x}_1, ..., \mathbf{x}_m \in \mathbb{R}^d$ and $W_1', ..., W_{2^m}' \in \mathbb{R}^{n \times d}$ be the vectors and metrices that are defined in Lemma 2. Note that $n = \Theta(e^d)$ and $m = \Theta(n)$.

Let $W_s = \frac{8\epsilon}{L} \cdot W_s'$ for each $s \in [2^m]$. By property 3 from that Lemma we have that $||W_s||_F^2 \leq 128d\epsilon^2/L^2$, for each $s \in [2^m]$. Thus, by setting $d := \lfloor \frac{L^2 B^2}{128\epsilon^2} \rfloor$, we get that $||W_s||_F \leq B$. Moreover, we have that $m = \Theta(n) = 2^{\Theta(L^2 B^2/\epsilon^2)}$. By property 4 from that lemma, we have that the set $Q = \{W_s \mathbf{x}_i : i \in [m], s \in [2^m]\}$ contains $m2^m$ different elements such that for each pair $W_s \mathbf{x}_i \neq W_t \mathbf{x}_j$ we have

$$||W_s \mathbf{x}_i - W_t \mathbf{x}_j|| = \frac{8\epsilon}{L} \cdot ||W_s' \mathbf{x}_i - W_t' \mathbf{x}_j|| \geq \frac{2\epsilon}{L}.$$

Order the elements of the set $2^{[m]}$ as $S_1, \ldots, S_{2^m}$ in some arbitrary order, and define the function $g : [m] \times [2^m] \to \{\pm\epsilon\}$ as:

$$g(k, i) = \begin{cases} \epsilon & i \in S_k \\ -\epsilon & i \notin S_k \end{cases}.$$

Apply Lemma 3 with the Euclidean metric space, to get an $L$-Lipschitz function, $f : \mathbb{R}^n \to \mathbb{R}$, such that for all $i \in [m], s \in [2^m]$,

$$f(W_s \mathbf{x}_i) = g(k, i).$$

Moreover, we have that $\mathbf{x} \to f(W_s \mathbf{x})$ belongs to $\mathcal{F}_{B,n,d}^{f, W_0 = 0}$ for each $s \in [2^m]$. Therefore, $\mathcal{F}_{B,n,d}^{f, W_0 = 0}$ can shatter $m = \Theta(n) = \exp\left(\Theta(L^2 B^2/\epsilon^2)\right)$ points from $\{\mathbf{x} \in \mathbb{R}^d : ||\mathbf{x}|| \leq 1\}$ with margin $\epsilon$. $\quad\square$

## A.3 Proof of Lemma 1

Define the $L_\infty$ distance

$$d_\infty(g, g') = \sup_{z \in \mathcal{Z}} \|f(z) - f'(z)\|,$$

where $\mathcal{Z}$ is the domain of $g$ and $g'$. Let $\mathcal{U}_B = \{\mathbf{x} \in \mathbb{R}^r : \|\mathbf{x}\| \leq B\}$ and fix some $\epsilon > 0$. By Observation 2 there exists a set $N_x \subseteq \mathbb{R}^r$ with $|N_x| = (1 + \frac{8BL}{\epsilon})^r$ s.t. for every $\mathbf{x} \in \mathcal{U}_B$, there exists $\mathbf{x}' \in N_x$ with

$$\|\mathbf{x} - \mathbf{x}'\| \leq \frac{\epsilon}{4L}.$$

Note that $\psi(\mathbf{x}) \in [a - BL, a + BL]$ for any $\psi \in \Psi_{L,a}, \mathbf{x} \in \mathcal{U}_B$. For simplicity, we assume that $a = 0$ (the proof for any other $a$ is essentially the same). Let $N_y := \{0, \pm\frac{\epsilon}{4}, \pm\frac{\epsilon}{2}, \ldots, \pm\lfloor BL/4\epsilon \rfloor 4\epsilon\}$ be $\frac{\epsilon}{4}$-cover for the closed interval $[-BL, BL]$. Given any $\psi \in \Psi_{L,a=0}$ and $\mathbf{x} \in \mathcal{U}_B$, construct $\psi' : N_x \to N_y$ as follows: let $\mathbf{x}'$ be the point in $N_x$ that is nearest to $\mathbf{x}$, and let $\psi'(\mathbf{x})$ be the point in $N_y$ that is nearest to $\psi(\mathbf{x}')$. We get that

$$|\psi(\mathbf{x}) - \psi'(\mathbf{x})| \leq |\psi(\mathbf{x}) - \psi(\mathbf{x}')| + |\psi(\mathbf{x}') - \psi'(\mathbf{x}')| + |\psi'(\mathbf{x}') - \psi'(\mathbf{x})|$$
$$\leq L \cdot \|\mathbf{x} - \mathbf{x}'\| + \frac{\epsilon}{4} + 0 \leq \frac{\epsilon}{2}.$$

The number of such functions is at most

$$|N_y|^{|N_x|},$$

therefore

$$\log N(\Psi_{L,a=0}, d_\infty, \frac{\epsilon}{2}) = |N_x| \log |N_y| = \left(1 + \frac{8BL}{\epsilon}\right)^r \cdot \log \frac{8BL}{\epsilon}.$$

Next, we argue that

$$\log N(\Psi_{L,a=0} \circ \mathcal{F}, d_m, \epsilon) \leq \log N(\Psi_{L,a=0}, d_\infty, \frac{\epsilon}{2}) + \log N(\mathcal{F}, d_m, \frac{\epsilon}{4L}),$$

from which the result will follow. To see this, pick any $\psi \in \Psi_{L,a=0}$ and $g \in \mathcal{F}$. By observation 1 there exists a proper $\frac{\epsilon}{2L}$-cover for $\mathcal{F}$ of size $N(\mathcal{F}, d_m, \frac{\epsilon}{4L})$. Let $\psi', g'$ be the respective closest functions in the cover of $\Psi_{L,a=0}$ and the proper cover of $\mathcal{F}$ (at scale $\frac{\epsilon}{2}$ and $\frac{\epsilon}{2L}$ respectively). Since $g'$ belongs to proper cover e.g. $g' \in \mathcal{F}$, its range is $\mathcal{U}_B$. By the triangle inequality and since $\psi$ is $L$-Lipschitz , we have

$$d_m(\psi g, \psi' g') \leq d_m(\psi g, \psi g') + d_m(\psi g', \psi' g') \leq d_m(\psi g, \psi g') + d_\infty(\psi g', \psi' g') \leq$$
$$L \cdot d_m(g, g') + d_\infty(\psi g', \psi' g') \leq \frac{\epsilon}{2} + \frac{\epsilon}{2} = \epsilon$$

## A.4 Proof of Theorem 2

**Observation 2.** *Let $\mathcal{U}_B := \{\mathbf{x} \in \mathbb{R}^r : \|\mathbf{x}\| \leq B\}$ and $\epsilon > 0$. There is a set $N \subseteq \mathcal{U}_B$ with $(B/\epsilon)^r \leq |N| \leq \left(1 + \frac{2B}{\epsilon}\right)^r$ such that:*

1. *For every $\mathbf{x} \in \mathcal{U}_B$, there exists a $\mathbf{x}' \in N$ with $\|\mathbf{x} - \mathbf{x}'\| \leq \epsilon$ (N is an $\epsilon$-cover for $\mathcal{U}_B$).*

2. *For every $\mathbf{x}, \mathbf{y} \in N$, we have that $\|\mathbf{x} - \mathbf{y}\| \geq \epsilon$ (N is an $\epsilon$-packing for $\mathcal{U}_B$).*

*Proof.* This is a well-known volume argument. Let $\mathcal{U}_B := \{\mathbf{x} \in \mathbb{R}^r : \|\mathbf{x}\| \leq B\}$ and fix some $\epsilon > 0$. Choose $N$ to be a maximal $\epsilon$-packing ($\epsilon$-seperated) subset of $\mathcal{U}_B$. In other words, $N \subseteq \mathcal{U}_B$ is such that $\|\mathbf{x} - \mathbf{y}\| \geq \epsilon$ for all $\mathbf{x}, \mathbf{y} \in N, \mathbf{x} \neq \mathbf{y}$ and no subset of $\mathcal{U}_B$ has this property. The maximality property implies that for every $\mathbf{x} \in \mathcal{U}_B$, there exists $\mathbf{x}' \in N$ with $\|\mathbf{x} - \mathbf{x}'\| \leq \epsilon$ (i.e. $N$ is an $\epsilon$-cover for $\mathcal{U}_B$). Otherwise there would exist $\mathbf{x} \in \mathcal{U}_B$ that is at least $\epsilon$-far from all points in $N$. Thus $N \cup \{\mathbf{x}\}$ would still be an $\epsilon$-separated set, contradicting the maximality property. Moreover, the separation property implies via the triangle inequality that the balls of radius $\frac{\epsilon}{2}$ centered at the points in $N$ are disjoint(up to null set). On the other hand, all such balls lie in $(B + \epsilon)B_2^r$ where $B_2^r$ denotes the unit Euclidean

ball centered at the origin. Comparing the volume gives $vol\left(\frac{\epsilon}{2}B_2^r\right) \cdot |N| \leq vol\left(\left(B+\frac{\epsilon}{2}\right)B_2^r\right)$ and $|N| \cdot vol(\epsilon B_2^r) \geq vol(B \cdot B_2^r)$. Since $vol(cB_2^r) = c^r vol(B_2^r)$ for all $c \geq 0$ we have

$$|N| \leq \frac{vol\left(\left(B+\frac{\epsilon}{2}\right)B_2^r\right)}{vol\left(\frac{\epsilon}{2}B_2^r\right)} = \left(\frac{B+\frac{\epsilon}{2}}{\frac{\epsilon}{2}}\right)^r,$$

$$|N| \geq \frac{vol(B \cdot B_2^r)}{vol(\epsilon \cdot B_2^r)} = \left(\frac{B}{\epsilon}\right)^r.$$

We conclude that $\left(\frac{B}{\epsilon}\right)^r \leq |N| \leq (1+\frac{2B}{\epsilon})^r$, as required. $\qquad\square$

The next lemma is shown in Corollary 9 in Kakade et al..

**Lemma 4.** *Let* $\mathcal{W} = \{\mathbf{x} \to \langle \mathbf{w}, \mathbf{x}\rangle : \|\mathbf{w}\| \leq B\}$ *be the class of norm-bounded linear predictors over inputs from* $\{\mathbf{x} : \|\mathbf{x}\| \leq 1\}$. *Then for every* $\epsilon > 0$

$$\log N(\mathcal{W}, d_m, \epsilon) \leq \frac{cB^2}{\epsilon^2},$$

*for some universal constant* $c > 0$.

**Lemma 5.** *Let*

$$\mathcal{F} = \{\mathbf{x} \to W\mathbf{x} : W \in \mathbb{R}^{r \times d}, \|W\|_F \leq B\}$$

*be class of functions over inputs from* $\{\mathbf{x} : \|\mathbf{x}\| \leq 1\}$. *Then for every* $\epsilon > 0$,

$$\log N(\mathcal{F}, d_m, \epsilon) \leq \frac{cr^2 B^2}{\epsilon^2},$$

*for some universal constant* $c > 0$.

*Proof.* Applying Lemma 4 with $\epsilon' = \frac{\epsilon}{\sqrt{r}}$, we get a class of functions $N_w$ with $|N_w| = 2^{crB^2/\epsilon^2}$ such that for every function $f_w \in \mathcal{W} := \{\mathbf{x} \to \mathbf{w}^\top \mathbf{x} : \|\mathbf{w}\| \leq B\}$, there exists $f'_w \in N_w$ with

$$d_m(f_w, f'_w) \leq \frac{\epsilon}{\sqrt{r}}.$$

Now we are ready to define a cover for $\mathcal{F}$. Let

$$N = \{(\mathbf{x}_1, ..., \mathbf{x}_r) \to (f'_1(\mathbf{x}_1), ..., f'_r(\mathbf{x}_r)) : f'_j \in N_w\}.$$

Let $f \in \mathcal{F}$ be $f(\mathbf{x}) = W\mathbf{x}$. Then

$$f(\mathbf{x}) = (\mathbf{w}_1^\top \mathbf{x}, ..., \mathbf{w}_r^\top \mathbf{x}),$$

where $\mathbf{w}_j$ is the $j$-th row of $W$. Define $f_j(\mathbf{x}) := \mathbf{w}_j^\top \mathbf{x}$. Since $\|W\|_F \leq B$, we have that $\|\mathbf{w}_j\| \leq B$, in particular $f_j \in \mathcal{W}$. Remember that $N_w$ is cover for $\mathcal{W}$, so there exists $f'_j \in N_w$ s.t.

$$d_m(f_j, f'_j) \leq \frac{\epsilon}{\sqrt{r}}.$$

Let $f' \in N$ be the function

$$f'(\mathbf{x}) := (f'_1(\mathbf{x}), ..., f'_r(\mathbf{x})).$$

Hence,

$$d_m(f, f')^2 = \frac{1}{m}\sum_{i=1}^m \|f(\mathbf{x}_i) - f'(\mathbf{x}_i)\|^2 = \frac{1}{m}\sum_{i=1}^m\sum_{j=1}^r \|f_j(\mathbf{x}_i) - f'_j(\mathbf{x}_i)\|^2 = \sum_{j=1}^r\sum_{i=1}^m \|f_j(\mathbf{x}_i) - f'_j(\mathbf{x}_i)\|^2$$

$$= \sum_{j=1}^r d_m(f_j, f'_j)^2 \leq \epsilon^2.$$

Therefore, $N$ is an $\epsilon$-cover for $\mathcal{F}$ with

$$|N| = |N_w|^r = \left(2^{\frac{crB^2}{\epsilon^2}}\right)^r = 2^{\frac{cr^2 B^2}{\epsilon^2}}$$

$\qquad\square$

*Proof of Theorem 2.* Assume for now that $L = 1$ and let $W \in \mathbb{R}^{n \times d}$ be matrix such that $||W||_F \leq B$. The singular value decomposition of $W$ is a factorization of the form $W = USV^\top$, where $U \in \mathbb{R}^{n \times n}$ and $V \in \mathbb{R}^{d \times d}$ are orthogonal matrices and $S \in \mathbb{R}^{n \times d}$ is a diagonal matrix with non-negative real numbers on the diagonal. We also can assume W.L.O.G that the upper-left submatrix of $S$ is of the form $diag(\sigma_1, ..., \sigma_{\min\{d,n\}})$ and that $\sigma_1 \geq \cdots \geq \sigma_{\min\{d,n\}}$. Let $\tilde{W}_\epsilon := U\tilde{S}_\epsilon V^\top$ be a low-rank approximation to $W$. where, $\tilde{S}_\epsilon$ is defined by setting all the elements in $S$ that are $\leq \epsilon$ to zero. Formally, let $r \in [\min\{n, d\}]$ be the largest number such that $\sigma_r > \epsilon$ (or 0 if no such number exists) and partition $U, S$ and $V$ as follows:

$$U = [\ U_1 \quad U_2\ ], S = \begin{bmatrix} S_1 & 0 \\ 0 & S_2 \end{bmatrix}, V = [\ V_1 \quad V_2\ ]$$

where $U_1 \in \mathbb{R}^{n \times r}, S_1 = diag(\sigma_1, ..., \sigma_r) \in \mathbb{R}^{r \times r}$ and $V_1 \in \mathbb{R}^{r \times d}$. We define

$$\tilde{S}_\epsilon := \begin{bmatrix} S_1 & 0 \\ 0 & 0 \end{bmatrix},$$

which means that $\tilde{W}_\epsilon = U_1 \tilde{S}_\epsilon V_1^\top$. The proof strategy consists of two parts: First, we argue that $\tilde{W}_\epsilon$ approximates $W$ in spectral norm. Second, we argue that $\tilde{W}_\epsilon$ has a low rank, so it is enough to find a small cover for the class of functions $\Psi_{L=1,a,r} \circ \mathcal{W}_{B,r}$ where $r := rank(\tilde{W}_\epsilon)$, to get a small cover for $\Psi_{L,a,n} \circ \mathcal{W}_{B,n}$. Details follow. Let $\mathbf{x} \in \mathbb{R}^d$ such that $||\mathbf{x}|| \leq 1$. Since $U$ is an orthogonal matrix, we have that $||U\mathbf{x}|| = ||\mathbf{x}|| \leq 1$. Moreover, since the spectral norm of a matrix is equal to the largest singular value

$$||S - \tilde{S}_\epsilon|| = ||S_2|| = \sigma_{r+1} \leq \epsilon.$$

Altogether we have

$$||W\mathbf{x} - \tilde{W}_\epsilon\mathbf{x}|| = ||USV^\top\mathbf{x} - U\tilde{S}_\epsilon V^\top\mathbf{x}|| = ||SV^\top\mathbf{x} - \tilde{S}_\epsilon V^\top\mathbf{x}|| \leq ||S - \tilde{S}_\epsilon|| \cdot ||V^\top\mathbf{x}|| \leq \epsilon$$

which means that $\tilde{W}_\epsilon$ indeed approximates $W$, in the sense of the spectral norm. Moreover,

$$B^2 \geq ||W||_F^2 = \sum_{i=1}^{\min\{d,n\}} \sigma_i^2 \geq \sum_{i=1}^{r} \sigma_i^2,$$

and since $\sigma_i^2 \geq \epsilon^2$ for each $i \leq r$, we have that $r \leq \frac{B^2}{\epsilon^2}$. Thus, we have that $Rank(\tilde{W}_\epsilon) = r \leq \frac{B^2}{\epsilon^2}$. If we approximate $W$ up to $\epsilon$, then we can argue that for each $f \in \Psi_{L=1,a,n}$,

$$|f(W\mathbf{x}) - f(\tilde{W}_\epsilon\mathbf{x})| \leq ||W\mathbf{x} - \tilde{W}_\epsilon\mathbf{x}|| \leq \epsilon.$$

Define

$$\tilde{\mathcal{W}}_\epsilon := \{\tilde{W}_\epsilon : ||W||_F \leq B, W \in \mathbb{R}^{n \times d}\},$$

then we have by the triangle inequality that

$$\log N(\Psi_{L=1,a,n} \circ \mathcal{W}, d_m, 2\epsilon) \leq \log N(\Psi_{L=1,a,n} \circ \tilde{\mathcal{W}}_\epsilon, d_m, \epsilon). \tag{2}$$

Therefore, we now turn to find a small cover for

$$\Psi_{L=1,a,n} \circ \tilde{\mathcal{W}}_\epsilon \subseteq$$
$$\left\{ \mathbf{x} \to f(W\mathbf{x}) : \mathbf{x} \in \mathbb{R}^d, W \in \mathbb{R}^{n \times d}, ||W||_F \leq B, Rank(W) \leq r, r = \frac{B^2}{\epsilon^2}, f \in \Psi_{L=1,a,n} \right\}.$$

Remember that by the singular value decomposition, if $Rank(W) = r$, then $W = USV^\top$ where $U \in \mathbb{R}^{n \times r}, S = diag(\sigma_1, ..., \sigma_r) \in \mathbb{R}^{r \times r}$ and $V \in \mathbb{R}^{r \times d}$. Also, $U$ and $V$ are orthogonal matrices, and $||S||_F \leq B$. Thus, the class from above is equal to

$$\left\{ \mathbf{x} \to f(USV\mathbf{x}) : U, V \text{ are orthogonal}, S = diag(\sigma_1, ..., \sigma_r), ||S||_F \leq B, r = \frac{B^2}{\epsilon^2}, f \in \Psi_{L=1,a,n} \right\}. \tag{3}$$

Now we want to get rid of $U$. Observe that
$$\{U\mathbf{x} \to f(U\mathbf{x}) : U \text{ is orthogonal}, f \in \Psi_{L=1,a,n}\} \subseteq \Psi_{L=1,a,r},$$
where we remind that $\Psi_{L=1,a,r}$ is the class of all 1-Lipschitz functions from $\{\mathbf{x} \in \mathbb{R}^r : ||\mathbf{x}|| \le B\}$ to $\mathbb{R}$, such that $f(\mathbf{0}) = a$ for some fixed $a \in \mathbb{R}$. Moreover,

$$\left\{\mathbf{x} \to SV^\top \mathbf{x} : V \text{ is orthogonal}, S = diag(\sigma_1, ..., \sigma_r), ||S||_F \le B, r = \frac{B^2}{\epsilon^2}\right\}$$

$$\subseteq \left\{\mathbf{x} \to (\sigma_1^\top \mathbf{v}_1^\top \mathbf{x}, ..., \sigma_r \mathbf{v}_r^\top \mathbf{x}) : \mathbf{v}_i \in \mathbb{R}^d, \sigma_i \in \mathbb{R}, ||\mathbf{v}_i|| = 1, \sum_{i=1}^r \sigma_i^2 = B^2, r = \frac{B^2}{\epsilon^2}\right\},$$

where $\mathbf{v}_i$ is the $i$-th column of $V$. Combining these observations, we have that the class of functions defined in Equation 3 is a subset of

$$\left\{\mathbf{x} \to f(\mathbf{w}_1^\top \mathbf{x}, ..., \mathbf{w}_r^\top \mathbf{x}) : f \in \Psi_{L=1,a,r}, \sum_{i=1}^r ||\mathbf{w}_i||^2 = B^2, r = \frac{B^2}{\epsilon^2}\right\} = \Psi_{L=1,a,r} \circ \mathcal{W}_{B,r}. \quad (4)$$

This class is a composition of all linear functions from $\mathbb{R}^d$ to $\mathbb{R}^r$ of Frobenius norm at most $B$, and all 1-Lipschitz functions. The covering number of such linear functions analyzed in Lemma 5, and the covering number of such composed classes analyzed in Lemma 1. Altogether, we have that the covering number of the class in Eq. 4 is upper bound by

$$\log N(\Psi_{L=1,a,r} \circ \mathcal{W}_{B,r}, d_m, \epsilon) \le \left(1 + \frac{8B}{\epsilon}\right)^r \cdot \log\left(\frac{8B}{\epsilon}\right) + \log N(\mathcal{W}_{B,r}, m, \frac{\epsilon}{4})$$

$$= \left(1 + \frac{8B}{\epsilon}\right)^r \cdot \log\left(\frac{8B}{\epsilon}\right) + \frac{cr^2 B^2}{\epsilon^2},$$

for some universal constant $c > 0$ and $r = \frac{B^2}{\epsilon^2}$. From this point, $c > 0$ represents some universal constant that may change from line to line. Combining with Eq. 2 and the assumption that $\frac{B}{\epsilon} \ge 1$, we have

$$\log N(\Psi_{L=1,a,n} \circ \mathcal{W}, d_m, \epsilon) \le \left(\frac{B}{\epsilon}\right)^{\frac{cB^2}{\epsilon^2}}.$$

Observe that $c \cdot \Psi_{L=1,a,n} = \Psi_{L=c,ca,n}$ for $c > 0$, and hence it is easy to verify that $N(\Psi_{L=1,a,n} \circ \mathcal{W}, d_m, \epsilon) = N(\Psi_{L=c,a,n} \circ \mathcal{W}, d_m, \epsilon/c)$. Therefore, for general $L$-Lipschitz functions we have

$$\log N(\Psi_{L,a,n} \circ \mathcal{W}, d_m, \epsilon) \le \left(\frac{LB}{\epsilon}\right)^{\frac{cL^2 B^2}{\epsilon^2}}.$$

To convert the upper bound on the covering number to an upper bound on the Rademacher complexity, we turn to use the Dudley integral covering number bound (see Srebro and Sridharan). In particular, since $g(\mathbf{x}) \le LB$ for each $g \in \Psi_{L,a,n} \circ \mathcal{W}_{B,n}$ and $\mathbf{x} \in \mathbb{R}^d$ with $||\mathbf{x}|| \le 1$, we have

$$R_m(\Psi_{L,a,n} \circ \mathcal{W}_{B,n}) \le \inf_{\epsilon \ge 0} \left\{4\epsilon + \frac{12}{\sqrt{m}} \int_\epsilon^{LB} \sqrt{\log N(\Psi_{L,a,n} \circ \mathcal{W}_{B,n}, d_m, \tau)} d\tau\right\} \le$$

$$\inf_{\epsilon \ge 0} \left\{4\epsilon + \frac{12LB}{\sqrt{m}} \sqrt{\log N(\Psi_{L,a,n} \circ \mathcal{W}_{B,n}, d_m, \epsilon)}\right\} \le \inf_{\epsilon \ge 0} \left\{\frac{\epsilon}{2} + \frac{12LB}{\sqrt{m}} \sqrt{\log N(\Psi_{L,a,n} \circ \mathcal{W}_{B,n}, d_m, \frac{\epsilon}{8})}\right\} \le$$

$$\inf_{\epsilon \in [0,1]} \left\{\frac{\epsilon}{2} + \frac{12LB}{\sqrt{m}} \sqrt{\left(\frac{LB}{\epsilon}\right)^{\frac{cL^2 B^2}{\epsilon^2}}}\right\}. \quad (5)$$

Moreover, there exists a universal constant $c' > 0$, that for any $\epsilon \in [0, 1]$, if $m \ge \left(\frac{LB}{\epsilon}\right)^{\frac{c' L^2 B^2}{\epsilon^2}}$, then

$$\frac{12LB}{\sqrt{m}} \sqrt{\left(\frac{LB}{\epsilon}\right)^{\frac{cL^2 B^2}{\epsilon^2}}} \le \frac{\epsilon}{2}.$$

Combining with Eq. 5, the Rademacher complexity of $\Psi_{L,a,n} \circ \mathcal{W}_{B,n}$ on $m$ inputs from $\{\mathbf{x} \in \mathbb{R}^d : \|\mathbf{x}\| \leq 1\}$ is at most $\epsilon$, if $m \geq \left(\frac{LB}{\epsilon}\right)^{\frac{c'L^2B^2}{\epsilon^2}}$. $\qquad\qquad\qquad\qquad\qquad\qquad\qquad\qquad\qquad\qquad\qquad\square$

## A.5   Proof of Theorem 3

Let $\mathbf{e}_i^d$ denote the indicator vector in $\mathbb{R}^d$ with value one in the $i$-th coordinate, and value zero in the other coordinates. If $d$ is clear from the context, we just use $\mathbf{e}_i$ for simplicity. Let $d = m + 1$ and $n = 2m$. Let $\mathbf{x}_1, \ldots, \mathbf{x}_m \in \{0,1\}^d$ be defined by

$$\mathbf{x}_i = \frac{1}{\sqrt{2}} \cdot (\mathbf{e}_i^d + \mathbf{e}_{m+1}^d).$$

Let $\epsilon \in (0, \frac{1}{4}]$ and define

$$W_0 = 2\sqrt{2} \cdot \epsilon \cdot \left[ \begin{array}{cc} I_{m \times m} & 0_{m \times 1} \\ & 0_{m \times (m+1)} \end{array} \right] \in \mathbb{R}^{n \times d}.$$

By observation 2 (part 2, with $B = 1, \epsilon = 1/2$), there exists $\mathbf{z}_1, \cdots, \mathbf{z}_{2^m} \in \mathbb{R}^m$ such that $\|z_i\| \leq 1, \|z_i - z_j\| \geq 1/2$ for any $i, j \in [2^m], i \neq j$. For any $\mathbf{y} \in \{\pm\epsilon\}^m$, we associate a different number from $[2^m]$ and we denote this number by $y$. For any $y \in [2^m]$ define

$$W_y' = \left[ \begin{array}{cc} 0_{n \times m} & \begin{array}{c} 0_{m \times 1} \\ \mathbf{z}_y \end{array} \end{array} \right] \in \mathbb{R}^{n \times d},$$

where $\mathbf{z}_y \in \{0,1\}^m$ is a column vector. Note that

$$W_0 \mathbf{x}_i = 2\epsilon \cdot \left[ \begin{array}{c} e_i^d \\ 0_{m \times 1} \end{array} \right] \in R^n, \quad W_y' \mathbf{x}_i = \frac{1}{\sqrt{2}} \cdot \left[ \begin{array}{c} 0_{m \times 1} \\ \mathbf{z}_y \end{array} \right] \in R^n.$$

Let $W_y = W_0 + W_y'$ for each $y \in [2^m]$, then

$$W_y \mathbf{x}_i = \left[ \begin{array}{c} 2\epsilon \cdot e_i^d \\ (1/\sqrt{2})\mathbf{z}_y \end{array} \right]$$

for all $i \in [m]$. Thus,

$$\|W_y \mathbf{x}_i - W_t \mathbf{x}_j\| = \left\| \left[ \begin{array}{c} 2\epsilon(e_i^d - e_j^d) \\ (1/\sqrt{2})(\mathbf{z}_y - \mathbf{z}_t) \end{array} \right] \right\| \geq 2\epsilon$$

for each $i, j \in [m]$ and $y, t \in [2^m]$ s.t. $(y, i) \neq (t, j)$. Note that the last inequality holds since $\epsilon < 1/4$. Apply Lemma 3 with the Euclidean metric space, on the set $Q = \{W_y \mathbf{x}_i : i \in [m], y \in [2^m]\}$ that contains $m2^m$ different elements, to get a 1-Lipchitz function $f : \mathbb{R}^n \to \mathbb{R}$ such that

$$f(W_y \mathbf{x}_i) = y_i \quad \forall i \in [m], y \in [2^m], \mathbf{y} \in \{\pm\epsilon\}^m$$

Moreover, note that since $\|W_y'\|_F \leq 1$, we have

$$\|W_y - W_0\|_F = \|W_y'\|_F \leq 1, \quad \|W_0\| = 2\sqrt{2} \cdot \epsilon$$

Namely, we have that the function $\mathbf{x} \to f(W_y \mathbf{x})$ belongs to $\mathcal{F}_{B=1,n,d}^{f,W_0}$ with $\|W_0\| = 2\sqrt{2} \cdot \epsilon$ for each $y \in [2^m]$. Therefore, $\mathcal{F}_{B=1,n,d}^{f,W_0}$ can shatter $m$ points from $\{\mathbf{x} \in \mathbb{R}^d : \|\mathbf{x}\| \leq 1\}$ with margin $\epsilon$.

## A.6   Proof of Theorem 4

This theorem is an adaption of Corollary 14.12 (page 197) in Shalev-Shwartz and Ben-David [2014]. The stochastic gradient descent (SGD) algorithm, with projection step and initialization at $W_0$, is describes as Algorithm 1 above. Observe that we just describe plain SGD on a convex Lipschitz stochastic optimization problem (over matrices, which is a vector space). The only nonstandard thing is the initialization at $W_0$ and the projection around $W_0$ instead of $\mathbf{0}$. We state the key technical result as Lemma 6 (in turn an adaptation of Lemma 14.1 from Shalev-Shwartz and Ben-David [2014]), and sketch its proof for completeness.

**Algorithm 1** Stochastic Gradient Descent (SGD), with projection step and initialization at $W_0$

---
**parameters:** Scalar $\eta > 0$, integer $T > 0$, vector $W_0$
**initialize:** $W^{(1)} = 0$
**for** $t = 1, 2, ..., T$ **do**
    Sample $(x_i, y_i)$
    pick a subgradient $V_t$ of $\ell(W_t x_i, y_i)$ w.r.t $W_t$
    update $W^{(t+\frac{1}{2})} = W^{(t)} - \eta V_t$
    update $W^{(t+1)} = \arg\min_{W:\|w-w_0\|_F \le B} \left\| W - W^{(t+\frac{1}{2})} \right\|_F$            ▷ Projection step
**end for**
**output:** $\hat{W} = \frac{1}{T} \sum_{t=1}^{\top} W^{(t)}$

---

**Lemma 6.** *Let $V_1, \ldots, V_T$ be an arbitrary sequence of matrices. Any algorithm with an initialization $W_1 = W_0$ and an update rule of the form*

- $W^{(t+\frac{1}{2})} = W^{(t)} - \eta V_t$

- $W^{(t+1)} = \arg\min_{W:\|W-W_0\|_F \le B} \left\| W - W^{(t+\frac{1}{2})} \right\|_F$

*satisfies*

$$\sum_{t=1}^{\top} \langle W^{(t)} - W^*, V_t \rangle \le \frac{\|W^* - W_0\|_F^2}{2\eta} + \frac{\eta}{2} \sum_{t=1}^{\top} \|V_t\|_F^2 \ ,$$

*where $\langle U, V \rangle = \sum_{i,j} U_{i,j} V_{i,j}$. In particular, for every $B, L > 0$, if for all $t$ we have that $\|V_t\|_F \le L$ and if we set $\eta = \sqrt{\frac{B^2}{L^2 T}}$, then for every $W^*$ with $\|W^* - W_0\|_F \le B$ we have*

$$\sum_{t=1}^{\top} \langle W^{(t)} - W^*, V_t \rangle \le \frac{BL}{\sqrt{T}}$$

**Proof sketch of Lemma 6:** Lemma 6 is different than Lemma 14.1 in Shalev-Shwartz and Ben-David [2014] in two aspects: first, we add a projection step, but still Lemma 14.1 holds (see Subsection 14.4.1 from Shalev-Shwartz and Ben-David [2014]). Second, the initialization is at $W_0$ and not 0, but this is also not a problem since that at the end of the proof of Lemma 14.1, Shalev-Shwartz and Ben-David [2014] showed that

$$\sum_{t=1}^{\top} \langle W^{(t)} - W^*, V_t \rangle \le \frac{1}{2\eta} \|W^{(1)} - W^*\|_F^2 + \frac{\eta}{2} \sum_{t=1}^{\top} \|V_t\|_F^2.$$

In our case $W^{(1)} = W_0$, this proves the first part of the Lemma. The second part follows by upper bounding $\|W_0 - W^*\|_F$ by B, $\|V_t\|_F$ by $L$ which is true since f is $L$-Lipschitz , dividing by T, and plugging in the value of $\eta$.

With Lemma 6 in hand, the rest of the analysis follows directly as in Corollary 14.12 Shalev-Shwartz and Ben-David [2014], only instead of Lemma 14.1 from that book, we use Lemma 6.

### A.7 Proof of Theorem 5

**Observation 3.** *Let $f_1, ..., f_k$ be L-Lipschitz functions from $\mathbb{R}^d$ to $\mathbb{R}$ with respect to the $L_p$ norm (with $p \in [1, \infty]$), then*

$$f(\mathbf{x}) := \max_{1 \le i \le k} f_i(\mathbf{x})$$

*is also an L-Lipschitz function with respect to the $L_p$ norm.*

*Proof.* First, we show a known property about $L_\infty$. If $\mathbf{x}, \mathbf{y} \in \mathbb{R}^k$, then

$$|\max_{1 \le i \le k} x_i - \max_{1 \le i \le k} y_i| \le \max_{1 \le i \le k} |x_i - y_i| \qquad (6)$$

Indeed, assume that $\max_i x_i > \max_i y_i$. In this case, we have

$$|\max_i x_i - \max_i y_i| = \max_i x_i - \max_i y_i$$

let us denote by $i_0$ the index $i_0 \in [k]$ such that $x_{i_0} = \max_i x_i$. Then we have

$$\max_i x_i - \max_i y_i = x_{i_0} - \max_i y_i \le x_{i_0} - y_{i_0} \le \max_i(x_i - y_i) \le \max_i |x_i - y_i|.$$

The case $\max_i y_i \ge \max_i x_i$ is symmetric. By Eq. 6 and the assumption that $f_i$ is $L$-Lipschitz for each $i \in [m]$ we get

$$|f(\mathbf{x}) - f(\mathbf{y})| = |\max_i f_i(\mathbf{x}) - \max_i f_i(\mathbf{y})| \le \max_i |f_i(\mathbf{x}) - f_i(\mathbf{y})| \le L||\mathbf{x} - \mathbf{y}||,$$

from which the result follows. $\qquad\square$

*Proof of Theorem 5.* Let $\mathbf{e}_i^d$ denote the indicator vector in $\mathbb{R}^d$ with value one in the $i$-th coordinate, and value zero in the other coordinates. If $d$ is clear from the context, we just use $\mathbf{e}_i$ for simplicity. Let $d = m + 1$ and $n = 2^m + m$. Let $\mathbf{x}_1, \ldots, \mathbf{x}_m \in \{0, 1\}^d$ be defined by $\mathbf{x}_i = \frac{1}{\sqrt{2}}(\mathbf{e}_i^d + \mathbf{e}_{m+1}^d)$. Define

$$W_0 = 4\sqrt{2} \cdot \epsilon \cdot \begin{bmatrix} I_{m \times m} & 0_{m \times 1} \\ 0_{2^m \times (m+1)} & \end{bmatrix} \in \mathbb{R}^{n \times d}$$

For any $\mathbf{y} \in \{\pm\epsilon\}^m$, we associate a different number from $[2^m]$, and we denote this number by $y$. For any $y \in [2^m]$ define

$$W'_y = \begin{bmatrix} 0_{n \times m} & \begin{matrix} 0_{m \times 1} \\ \mathbf{e}_y \end{matrix} \end{bmatrix}$$

where $\mathbf{e}_y \in \{0, 1\}^{2^m}$ is a column vector. Note that $W_0 \mathbf{x}_i = 4\epsilon \cdot \mathbf{e}_i^n$ and $W'_y \mathbf{x}_i = \frac{1}{\sqrt{2}}\mathbf{e}_{y+m}^n$. Letting $W_y = W_0 + W'_y$, we have that

$$W_y \mathbf{x}_i = 4\epsilon \mathbf{e}_i^n + \frac{1}{\sqrt{2}}\mathbf{e}_{m+y}^n.$$

Now we are ready to define $f : \mathbb{R}^n \to \mathbb{R}$,

$$f(\mathbf{x}) := \max_{j \in [m], z \in [2^m] \text{ s.t.} \mathbf{z}_j = \epsilon} \left[ (0.5 \cdot \mathbf{e}_j^n + 0.5 \cdot \mathbf{e}_{m+z}^n)^\top \mathbf{x}, \frac{1}{\sqrt{8}} \right] - (\frac{1}{\sqrt{8}} + \epsilon),$$

for any $j \in [m], z \in [2^m]$ and $\mathbf{x} \in \mathbb{R}^n$. We emphasize that $\mathbf{z} \in \{\pm\epsilon\}^m$ is the vector that is associated with the number $z \in [2^m]$. By Hölder's inequality, we have that

$$|(0.5 \cdot \mathbf{e}_j^n + 0.5 \cdot \mathbf{e}_{m+z}^n)^\top (\mathbf{x} - \mathbf{y})| \le ||0.5 \cdot \mathbf{e}_j^n + 0.5 \cdot \mathbf{e}_{m+z}^n||_1 \cdot ||\mathbf{x} - \mathbf{y}||_\infty \le ||\mathbf{x} - \mathbf{y}||_\infty,$$

for any $\mathbf{x}, \mathbf{y} \in \mathbb{R}^n$. Therefore, the function $\mathbf{x} \to (0.5 \cdot \mathbf{e}_j^n + 0.5 \cdot \mathbf{e}_{m+z}^n)^\top \mathbf{x}$ is 1-Lipschitz with respect to the infinity norm for each $j \in [m], z \in [2^m]$. This implies by Observation 3 that $f$ is also a 1-Lipchitz function with respect to the infinity norm. Since the composition of an affine map, nonnegative weighted sums, maximum, and adding a constant are all operations that preserve convexity, we have that $f$ is a convex function. Finally, for any $\mathbf{y} \in \{\pm\epsilon\}^m$ and $\mathbf{x}_i$ we have

- If $y_i = \epsilon$, then

$$\begin{aligned} f(W_y \mathbf{x}_i) &= f(4\epsilon \mathbf{e}_i^n + (1/\sqrt{2})\mathbf{e}_{m+y}^n) \\ &= \max_{j \in [m], z \in [2^m] \text{ s.t.} \mathbf{z}_j = \epsilon} (0.5\mathbf{e}_j^n + 0.5\mathbf{e}_{m+z}^n)^\top (4\epsilon \mathbf{e}_i^n + \mathbf{e}_{y+m}^n) - (1/\sqrt{8} + \epsilon) \\ &= (0.5\mathbf{e}_i^n + 0.5\mathbf{e}_{m+y}^n)^\top (4\epsilon \mathbf{e}_i^n + (1/\sqrt{2})\mathbf{e}_{m+y}^n) - (1/\sqrt{8} + \epsilon) = \epsilon. \end{aligned}$$

- If $y_i = -\epsilon$ and exists $k \in [m]$ with $y_k = \epsilon$, then

$$f(W_y\mathbf{x}_i) = f(4\epsilon\mathbf{e}_i^n + (1/\sqrt{2})\mathbf{e}_{m+y}^n)$$

$$= \max_{j \in [m], z \in [2^m] \text{ s.t.} \mathbf{z}_j = \epsilon} (0.5\mathbf{e}_j^n + 0.5\mathbf{e}_{m+z}^n)^\top(4\epsilon\mathbf{e}_i^n + (1/\sqrt{2})\mathbf{e}_{m+y}^n) - (1/\sqrt{8} + \epsilon)$$

$$= (0.5\mathbf{e}_k^n + 0.5\mathbf{e}_{m+y}^n)^\top(4\epsilon\mathbf{e}_i^n + (1/\sqrt{2})\mathbf{e}_{m+y}^n) - (1/\sqrt{8} + \epsilon) = -\epsilon,$$

where in this case, the max is obtained for some $k \in [m]$ with $\mathbf{y}_k = \epsilon$.

- Otherwise, for every $k$ we have $y_k = -\epsilon$, then

$$f(W_y\mathbf{x}_i) = f(4\epsilon\mathbf{e}_i^n + (1/\sqrt{2})\mathbf{e}_{m+y}^n) = 1/\sqrt{8} - (1/\sqrt{8} + \epsilon) = -\epsilon.$$

In all cases, we get that

$$f(W_y\mathbf{x}_i) = y_i.$$

Therefore, $\mathcal{F}_{B=1,n,d}^{f,W_0}$ with $\|W_0\| = \sqrt{2} \cdot 4\epsilon$ can shatter $m$ points from $\{\mathbf{x} \in \mathbb{R}^d : \|\mathbf{x}\| \leq 1\}$ with margin $\epsilon$. $\qquad\square$

## A.8 Proof of Theorem 6

**Lemma 7.** *Let $\sigma : \mathbb{R} \to \mathbb{R}$ be L-Lipschitz and $\mu$-smooth. Let $\psi : I \to R$ (when $I := (I_1, I_2, I_3) \subseteq \mathbb{R}^3$) be the function $(x, y, v) \to \frac{\sigma(vx+y)-\sigma(y)}{v}$. If $I_1 = [-b_x, b_x]$ and $I_3 = (0, \infty)$, then $\psi$ is $O(\mu b_x^2 + L)$-Lipschitz.*

*Proof.* We show that $\psi$ is Lipschitz in each coordinate, which implies that $\psi$ is Lipschitz. To that end, it is enough to upper bound the norm of the gradient:

$$\|\nabla\psi\| = \sqrt{\left(\frac{d\psi}{dx}\right)^2 + \left(\frac{d\psi}{dy}\right)^2 + \left(\frac{d\psi}{dv}\right)^2} \leq O\left(\left|\frac{d\psi}{dx}\right| + \left|\frac{d\psi}{dx}\right| + \left|\frac{d\psi}{dx}\right|\right).$$

Indeed,

$$\left|\frac{d\psi}{dx}\right| = \left|\frac{v\sigma'(vx+y)}{v}\right| \leq L.$$

Since $\sigma(\cdot)$ is $\mu$-smooth, namely $\sigma'(\cdot)$ is $\mu$-Lipschitz we have

$$\left|\frac{d\psi}{dy}\right| = \left|\frac{\sigma'(vx+y) - \sigma'(y)}{v}\right| \leq \left|\mu\frac{vx}{v}\right| = \mu|x| \leq \mu b_x.$$

Moreover,

$$\left|\frac{d\psi}{dv}\right| = \left|\frac{vx\sigma'(vx+y) - (\sigma(vx+y) - \sigma(y))}{v^2}\right| = \left|\frac{\sigma(y) - (\sigma(vx+y) + \sigma'(vx+y)(-vx))}{v^2}\right|.$$

Note that $\sigma(y) - (\sigma(vx+y) + \sigma'(vx+y)(-vx)$ is exactly the reminder between $\sigma(y)$ and the first-order Taylor expansion of $\sigma(y)$ at $vx + y$. In Observation 4 we analyzed such reminder of a smooth function and thus we can upper bound the above equation by

$$\frac{\mu}{2}\frac{(vx)^2}{v^2} \leq \frac{\mu}{2}b_x^2,$$

where c is a middle point between $y$ and $vx + y$. Therefore, $\psi$ is a $O(\mu b_x^2 + L)$-Lipchitz function, as required. $\qquad\square$

**Observation 4.** *Let $\sigma : \mathbb{R} \to \mathbb{R}$ be a continuously differentiable function. For all $x, y \in \mathbb{R}$, the first-order Taylor expansion of $\sigma(y)$ at $x$ is define by $\sigma(x) + \sigma'(x)(y - x)$, and the reminder $R_x(y)$ is define by*

$$\sigma(y) = \sigma(x) + \sigma'(x)(y - x) + R_x(y).$$

*If $\sigma(\cdot)$ is a $\mu$-smooth i.e. $\sigma'(\cdot)$ is an L-Lipschitz function, then*

$$|R_x(y)| \leq \frac{\mu}{2}|y - x|^2$$

*Proof.* Since $\sigma'(\cdot)$ is continuous, we have that

$$\int_0^1 \left(\sigma'(x + t(y - x)) - \sigma'(x)\right)(y - x)dt = \sigma(y) - \sigma(x) - \sigma'(x)(y - x),$$

then we can write

$$\sigma(y) = \sigma(x) + \sigma'(x)(y - x) + \int_0^1 \left(\sigma'(x + t(y - x)) - \sigma'(x)\right)(y - x)dt.$$

Since $\sigma(\cdot)$ is a $\mu$-smooth

$$|R_x(y)| = \left|\int_0^1 \left(\sigma'(x + t(y - x)) - \sigma'(x)\right)(y - x)dt\right| \leq$$

$$\int_0^1 |(\sigma'(x + t(y - x)) - \sigma'(x))(y - x)|\, dt \leq \mu|y - x|^2 \int_0^1 tdt = \frac{\mu}{2}|y - x|^2$$

$\square$

**Lemma 8.** *Let $\psi : \mathbb{R}^k \to R$ be an L-Lipschitz function. Namely for all $\alpha, \beta \in \mathbb{R}^k$ we have $|\psi(\alpha) - \psi(\beta)| \leq L||\alpha - \beta||$. For $f_1, ..., f_k$ functions from $\mathbb{R}^d$ to $\mathbb{R}$ and $\mathbf{x} \in \mathbb{R}^d$, let $\psi \circ (f_1, ..., f_k)(\mathbf{x}) := \psi(f_1(\mathbf{x}), ..., f_k(\mathbf{x}))$. For $\mathcal{F}_1, \ldots, \mathcal{F}_k$ class of functions from $\mathbb{R}^d$ to $\mathbb{R}$, let*

$$\psi \circ (F_1, ..., F_k) = \{\mathbf{x} \to \psi \circ (f_1, ..., f_k)(\mathbf{x}) : f_1 \in \mathcal{F}_1 \wedge ... \wedge f_k \in \mathcal{F}_k\}.$$

*Then,*

$$\log N(\psi \circ (F_1, ..., F_k), d_m, \sqrt{k}Lr) \leq \log N(F_1, d_m, r) + ... + \log N(F_k, d_m, r)$$

*Proof.* Define $B = \psi \circ (\mathcal{F}_1, ..., \mathcal{F}_k)$. Let $\mathcal{F}_1', ..., \mathcal{F}_k'$ be an $r$-covers of $\mathcal{F}_1, ..., \mathcal{F}_k$ respectively. Define $B' = \psi \circ (\mathcal{F}_1', ..., \mathcal{F}_m')$. For all $f_1 \in \mathcal{F}_1, ..., f_k \in \mathcal{F}_k$, there exists $f_1' \in \mathcal{F}_1', ..., f_k' \in \mathcal{F}_k'$ s.t.

$$d_m(f_i, f_i') \leq r$$

For $i = 1, ..., k$. Therefore,

$$d_m(\psi(f_1, ..., f_k), \psi(f_1', ..., f_k')) = \frac{1}{m} \sum_{i=1}^m \left(\psi \circ (f_1, ..., f_k)(\mathbf{x}_i) - \psi \circ (f_1', ..., f_k')(\mathbf{x}_i)\right)^2 =$$

$$\leq \frac{L^2}{m} \sum_{i=1}^m \left\|(f_1(\mathbf{x}_i), ..., f_k(\mathbf{x}_i))^\top - (f_1'(\mathbf{x}_i), ..., f_k'(\mathbf{x}_i))^\top\right\|^2$$

$$= \frac{L^2}{m} \sum_{i=1}^m (f_1(\mathbf{x}_i) - f_1'(\mathbf{x}_i))^2 + ... + \frac{L^2}{m} \sum_{i=1}^m (f_k(\mathbf{x}_i) - f_k'(\mathbf{x}_i))^2$$

$$= L^2 d_m(f_1, f_1')^2 + \cdots + L^2 d_m(f_k, f_k')^2 \leq kL^2r^2.$$

Hence, $B'$ is an $(\sqrt{k}Lr) - cover$ for $B$. Moreover, since $|B'| \leq |\mathcal{F}_1'| \cdot ... \cdot |\mathcal{F}_k'|$, we have

$$N(\psi \circ (\mathcal{F}_1, ..., \mathcal{F}_k), d_m, \sqrt{k}Lr) \leq N(\mathcal{F}_1, d_m, r) \cdot ... \cdot N(\mathcal{F}_k, d_m, r)$$

$\square$

**Lemma 9.** *Let $\mathcal{F} = \{\mathbf{x} \to \langle \mathbf{w}, \mathbf{x} \rangle : ||\mathbf{w}|| \leq B\}$ be the class of Euclidean norm-bounded linear predictors. Let $L > 0$ and $k > 0$ be some parameters, then for every $\epsilon \in (0, L]$,*

*1.* $\sqrt{\log N(\mathcal{F}, d_m, \epsilon)} \leq \frac{cBb_{\mathbf{x}}}{\epsilon}$

*2.* $\int_\epsilon^L \sqrt{\log N(\mathcal{F}, d_m, k\tau)}d\tau \leq \frac{cBb_x}{k} (\log(L) - \log(\epsilon))$

*For some universal constant $c > 0$.*

*Proof.* The first part of the lemma is shown in Corollary 9 in Kakade et al.. For the second part,

$$\int_\epsilon^L \sqrt{N(\mathcal{F}, d_m, k\tau)} d\tau \leq \int_\epsilon^L \frac{cBb_x}{k\tau} d\tau = \frac{cBb_x}{k} \left(\log(L) - \log(\epsilon)\right)$$

$\square$

**Lemma 10.** *Let $B \geq 2$ and let $F = \{\mathbf{x} \to v : v \in (0, B]\}$ be the class of constant functions. Let $L > 0$ and $k > 0$ be some parameters, then for every $\epsilon \in (0, L]$,*

1. $\log N(\mathcal{F}, d_m, \epsilon) \leq \frac{2\log_2(B)}{\epsilon}$

2. $\int_\epsilon^L \sqrt{\log N(\mathcal{F}, d_m, k\tau)} d\tau \leq \frac{2\log_2(B)}{k} \left(\log(L) - \log(\epsilon)\right)$

*Proof.* Using $n$ numbers we can represent every number in $[0, B]$ with an accuracy of $\epsilon = \frac{B}{n}$. Thus, $N(\mathcal{F}, d_m, \frac{B}{n}) \leq n$, namely $\log N(\mathcal{F}, d_m, \epsilon) \leq \log(\lceil \frac{B}{\epsilon} \rceil) \leq \frac{2\log(B)}{\epsilon}$ for each $0 < \epsilon \leq 1$ and $B \geq 2$, which proves the first part of the lemma. Moreover,

$$\int_\epsilon^L \sqrt{\log N(\mathcal{F}, d_m, k\tau)} d\tau \leq \int_\epsilon^L \frac{2\log(B)}{k\tau} d\tau = \frac{2\log(B)}{k} \left(\log(L) - \log(\epsilon)\right)$$

$\square$

*Proof of Theorem 6.* Fix some set of inputs $\mathbf{x}_1, ..., \mathbf{x}_m$ with norm at most $b_x$. The Rademacher complexity equals

$$\mathbb{E} \sup_{||W||_F \leq B} \frac{1}{m} \sum_{i=1}^m \epsilon_i v^\top \sigma\left((W + w_0)\mathbf{x}_i\right) \leq \frac{b}{m} \mathbb{E} \sup_{||W||_F \leq B} \left\| \sum_{i=1}^m \epsilon_i \sigma(W\mathbf{x}_i + W_0\mathbf{x}_i) \right\|$$

$$\leq \frac{b}{m} \mathbb{E} \sup_W \left\| \sum_{i=1}^m \epsilon_i \sigma(W\mathbf{x}_i + W_0\mathbf{x}_i) - \sum_{i=1}^m \epsilon_i \sigma(W_0\mathbf{x}_i) \right\| + \frac{b}{m} \mathbb{E} \left\| \sum_{i=1}^m \epsilon_i \sigma(W_0\mathbf{x}_i) \right\|. \tag{7}$$

Let's start by upper bound the right-hand side of Equation 7, namely

$$\frac{b}{m} \mathbb{E} \left\| \sum_{i=1}^m \epsilon_i \sigma(W_0\mathbf{x}_i) \right\|.$$

By definition of the spectral norm, we have that $\|W_0\mathbf{x}_i\| \leq B_0 b_x$. Since $\sigma(\cdot)$ is $L$-Lipschitz and $\sigma(0) = 0$ we have that $\|\sigma(W_0\mathbf{x}_i)\| \leq L B_0 b_x$. Let $y_i := \sigma(W_0\mathbf{x}_i)$ where $\|y_i\| \leq L B_0 b_x$. Then the expression above equals

$$\frac{b}{m} \mathbb{E} \left\| \sum_{i=1}^m \epsilon_i y_i \right\| = \frac{b}{m} \mathbb{E} \sqrt{\left\| \sum_{i=1}^m \epsilon_i y_i \right\|^2} \leq \frac{b}{m} \sqrt{\mathbb{E} \left\| \sum_{i=1}^m \epsilon_i y_i \right\|^2} = \sqrt{\sum_{j=1}^n \mathbb{E} \left( \sum_{i=1}^m \epsilon_i y_{i,j} \right)^2}$$

$$\overset{\mathbb{E}[\epsilon_i \epsilon_{i'}]=0}{=} \frac{b}{m} \sqrt{\sum_{i,j} y_{i,j}^2} = \frac{b}{m} \sqrt{\sum_{i=1}^m \|y_i\|^2} \leq \frac{L B_0 b b_x}{\sqrt{m}}. \tag{8}$$

Moving back to the left-hand side of Equation 7, let $\bar{\mathbf{x}} := \mathbf{x}/||\mathbf{x}||$ for any non-zero $\mathbf{x}$ (or 0 for $\mathbf{x} = \mathbf{0}$). We have

$$\frac{b}{m} \mathbb{E} \sup_{||W||_F \leq B} \left\| \sum_{i=1}^m \epsilon_i \sigma(W\mathbf{x}_i + W_0\mathbf{x}_i) - \sum_{i=1}^m \epsilon_i \sigma(W_0\mathbf{x}_i) \right\|$$

$$\leq \frac{b}{m} \mathbb{E} \sup_W \sqrt{\sum_{j=1}^n \left( \sum_{i=1}^m \epsilon_i \left( \sigma(b_x \mathbf{w}_j^\top \bar{\mathbf{x}}_i + b_x \mathbf{w}_{0,j}^\top \bar{\mathbf{x}}_i) - \sigma(b_x \mathbf{w}_{0,j}^\top \bar{\mathbf{x}}_i) \right) \right)^2},$$

where $\mathbf{w}_{0,j}$ is the j row of $W_0$.

Each matrix in the set $\{W : \|W\|_F \leq B\}$ is composed of rows, whose sum of squared norms is at most $(Bb_x)^2$. Thus, the set can be equivalently defined as the set of $d \times n$ matrices, where each row j equals $v_j \mathbf{w}_j$ for some $v_j > 0$, $\|w_j\| \leq 1$ and $\|\mathbf{v}\|^2 \leq (Bb_x)^2$. Noting that each $v_j$ is positive, we can upper bound the expression in the displayed equation above as follows:

$$\frac{b}{m} \mathbb{E} \sup_{\|\mathbf{v}\| \leq Bb_x, \|\mathbf{w}_j\| \leq 1} \sqrt{\sum_{j=1}^{n} \left( \sum_{i=1}^{m} \epsilon_i \left( \sigma(v_j \mathbf{w}_j^\top \bar{\mathbf{x}}_i + b_x \mathbf{w}_{0,j}^\top \bar{\mathbf{x}}_i) - \sigma(b_x \mathbf{w}_{0,j}^\top \bar{\mathbf{x}}_i) \right) \right)^2}$$

$$= \frac{b}{m} \mathbb{E} \sup_{\|\mathbf{v}\| \leq Bb_x, \|\mathbf{w}_j\| \leq 1} \sqrt{\sum_{j=1}^{n} v_j^2 \left( \sum_{i=1}^{m} \frac{\epsilon_i}{v_j} \left( \sigma(v_j \mathbf{w}_j^\top \bar{\mathbf{x}}_i + b_x \mathbf{w}_{0,j}^\top \bar{\mathbf{x}}_i) - \sigma(b_x \mathbf{w}_{0,j}^\top \bar{\mathbf{x}}_i) \right) \right)^2}$$

$$\leq \frac{b}{m} \mathbb{E} \sup_{\|\mathbf{v}'\| \leq B, \|\mathbf{v}\| \leq Bb_x, \|\mathbf{w}_j\| \leq 1} \sqrt{\sum_{j=1}^{n} v_j'^2 \left( \sum_{i=1}^{m} \frac{\epsilon_i}{v_j} \left( \sigma(v_j \mathbf{w}_j^\top \bar{\mathbf{x}}_i + b_x \mathbf{w}_{0,j}^\top \bar{\mathbf{x}}_i) - \sigma(b_x \mathbf{w}_{0,j}^\top \bar{\mathbf{x}}_i) \right) \right)^2}.$$

For any choice of $\epsilon, \mathbf{v}$ and $\mathbf{w}_1, ..., \mathbf{w}_n$, the expression inside the expectation above can be written as

$$\sup_{\|\mathbf{v}'\| \leq Bb_x} \sqrt{\sum_{j=1}^{n} v_j'^2 a_j^2} = \sup_{v_j' : \sum_j v_j'^2 \leq (Bb_x)^2} \sqrt{\sum_{j=1}^{n} v_j'^2 a_j^2},$$

for some numbers $a_1, ..., a_n$. Clearly, this is maximized by letting $v'_{j^*} = Bb_x$ for some $j^* \in \arg\max_j a_j^2$, and $v_j' = 0$ for all $j \neq j^*$. Plugging this observation back into the above expression, we can upper-bound the displayed equation by

$$\frac{bBb_x}{m} \mathbb{E} \sup_{\|\mathbf{v}\| \leq Bb_x, \|\mathbf{w}_j\| \leq 1} \max_j \left| \sum_{i=1}^{m} \frac{\epsilon_i}{v_j} \left( \sigma(v_j \mathbf{w}_j^\top \bar{\mathbf{x}}_i + b_x \mathbf{w}_{0,j}^\top \bar{\mathbf{x}}_i) - \sigma(b_x \mathbf{w}_{0,j}^\top \bar{\mathbf{x}}_i) \right) \right|.$$

Since the spectral norm upper bounds the norm of each row in a matrix, we can upper bound the above equation by

$$\leq \frac{bBb_x}{m} \mathbb{E} \sup_{v \in (0, Bb_x], \|\mathbf{w}\| \leq 1, \|w_0\| \leq B_0} \left| \sum_{i=1}^{m} \frac{\epsilon_i}{v} \left( \sigma(v\mathbf{w}^\top \bar{\mathbf{x}}_i + b_x \mathbf{w}_0^\top \bar{\mathbf{x}}_i) - \sigma(b_x \mathbf{w}_0^\top \bar{x}_i) \right) \right|.$$

Let $\psi : I \to \mathbb{R}$ be

$$(\alpha, y, v) \to \frac{\sigma(v\alpha + y) - \sigma(y)}{v},$$

where $I = [-1, 1] \times [-B_0 b_x, B_0 b_x] \times [-Bb_x, Bb_x]$. Note that $|\mathbf{w}^\top \bar{\mathbf{x}}_i| \leq \|\mathbf{w}^\top\|\|\bar{\mathbf{x}}_i\| \leq 1$ and $|b_x w_0^\top \bar{\mathbf{x}}_i| \leq \|\mathbf{w}_0^\top\|\|\bar{\mathbf{x}}_i\| \leq B_0 b_x$. Therefore we can upper-bound the above expression by

$$\frac{bBb_x}{m} \mathbb{E} \sup_{v \in (0, Bb_x], \|\mathbf{w}\| \leq 1, \|\mathbf{w}_0\| \leq B_0} \left| \sum_{i=1}^{m} \epsilon_i \psi(\mathbf{w}^\top \bar{\mathbf{x}}_i, \mathbf{w}_0^\top \mathbf{x}_i, v) \right|. \tag{9}$$

By Lemma 7 we have that $\psi$ is $c(\mu + L)$-Lipschitz for some constant c. By a Taylor expansion and the fact that $\sigma(\cdot)$ is smooth we have that $\sigma(\beta) = \sigma(v\alpha + \beta) + \sigma'(\gamma)(-v\alpha)$, where $\gamma$ is some middle point between $\beta$ and $v\alpha + \beta$. Therefore,

$$\psi(\alpha, \beta, v) = \frac{\sigma(v\alpha + \beta) - \sigma(\beta)}{v} = \frac{\sigma'(\gamma)(-v\alpha)}{v} = -\alpha\sigma'(\gamma).$$

since $|\alpha| \leq 1$ and $|\sigma'(\gamma)| \leq L$, we have that $\psi$ is bounded on $I$ by $L$. Let

$$\mathcal{F} = \{\mathbf{x} \to (\mathbf{w}^\top \bar{\mathbf{x}}, \mathbf{w}_0^\top \mathbf{x}, v) : \|\mathbf{w}\| \leq 1, \|\mathbf{w}_0\| \leq B_0, \|\mathbf{v}\| \leq Bb_x\},$$
$$\mathcal{F}_1 = \{\mathbf{x} \to \mathbf{w}^\top \bar{\mathbf{x}} : \|\mathbf{w}\| \leq 1\},$$
$$\mathcal{F}_2 = \{\mathbf{x} \to \mathbf{w}_0^\top \mathbf{x} : \|\mathbf{w}_0\| \leq B_0\},$$
$$\mathcal{F}_3 = \{\mathbf{x} \to v : v \in (0, Bb_x]\}.$$

Considering Equation 9, this is $bBb_x$ times the Rademacher complexity of the function class $\psi \circ \mathcal{F}$. For every $\epsilon > 0$, using the Dudley Integral (see Srebro and Sridharan) we can upper bound the Rademacher complexity of $\psi \circ \mathcal{F}$ by

$$4\epsilon + 12 \int_\epsilon^L \sqrt{\frac{\log N(\psi \circ F, d_m, \tau)}{m}} d\tau.$$

From this point, $c > 0$ represents some universal constant that may change from line to line. By Lemma 8 (with $k = 3$) we can upper bound the above expression by

$$4\epsilon + 12 \int_\epsilon^L \sqrt{\frac{\log N(F_1, d_m, \frac{c\tau}{\mu+L})}{m}} d\tau + 12 \int_\epsilon^L \sqrt{\frac{\log N(F_2, d_m, \frac{c\tau}{\mu+L})}{m}} d\tau + 12 \int_\epsilon^L \sqrt{\frac{\log N(F_3, d_m, \frac{c\tau}{\mu+L})}{m}} d\tau.$$

By Lemma 9 and Lemma 10, for any $\epsilon \geq 0$, we can upper bound the above expression by

$$4\epsilon + \frac{c(\mu+L)}{\sqrt{m}} \Big( \log(L) - \log(\epsilon) + B_0 b_x \log(L) - B_0 b_x \log(\epsilon) + \log(Bb_x) \log(L) - \log(Bb_x) \log(\epsilon) \Big).$$

By choosing $\epsilon = \frac{1}{\sqrt{m}}$ we can upper bound Eq. 9 by

$$\frac{4 + c(\mu+L)\big(\log(L) + \log(m) + B_0 b_x \log(L) + B_0 b_x \log(m) + \log(Bb_x)\log(L) + \log(Bb_x)\log(m)\big)}{\sqrt{m}}.$$

Combining with Eq. 8, we get an upper bound on the Rademacher Complexity of $\mathcal{F}_{b,B,n,d}^{g,W_0}$ of the form

$$\frac{4 + c(\mu+L)bBb_x\big(\log(L) + \log(m) + B_0 b_x \log(L) + B_0 b_x \log(m)\big)}{\sqrt{m}}$$
$$+ \frac{c(\mu+L)bBb_x\big(\log(Bb_x)\log(L) + \log(Bb_x)\log(m)\big)}{\sqrt{m}} + \frac{LB_0 bb_x}{\sqrt{m}}.$$

Upper bounding this by $\epsilon$ and solving for $m$, the result follows. □

### A.9 Proof of Theorem 7

To simplify notation, we rewrite $\mathcal{F}_{k,\{S_j\},\{B_j\}}^{\{\sigma_j\}}$ as simply $\mathcal{F}_k$. For convenience, for any $1 \leq l \leq k - 1$, we define the class $\mathcal{F}_l$ slightly differently. Each function in $\mathcal{F}_l$ has the form

$$\mathbf{x} \rightarrow \sigma_l(W_l \sigma_{l-1}(...\sigma_1((W_1 \mathbf{x})))), \tag{10}$$

with the same constraints on the weights and the activation functions. We also define $\mathcal{F}_0$ to be the class that contains just the identity function. The next Claim captures the "peeling" argument.

**Lemma 11.** *For any integer* $1 \leq l \leq k - 1$,

$$\mathbb{E} \sup_{f \in F_l} \left\| \frac{1}{m} \sum_{i=1}^m \epsilon_i f(\mathbf{x}_i) \right\| \leq 2cB_l L \left( \frac{R_{l-1}}{\sqrt{m}} + \log^{\frac{3}{2}}(m) \mathbb{E} \sup_{f \in F_{l-1}} \left\| \frac{1}{m} \sum_{i=1}^m \epsilon_i f(\mathbf{x}_i) \right\| \right)$$

*where* $R_{l-1} = b_x L^{l-1} ||W_{l-1}|| \cdot ||W_{l-2}|| \cdot \cdots \cdot ||W_1||$, $R_0 = b_x$ *and* $c > 0$ *is a universal constant.*

*Proof.*

$$\mathbb{E} \sup_{f \in F_l} \left\| \frac{1}{m} \sum_{i=1}^m \epsilon_i f(\mathbf{x}_i) \right\| = \frac{1}{m} \mathbb{E} \sup_{f \in \mathcal{F}_{l-1}} \sup_{W_l} \left\| \sum_{i=1}^m \epsilon_i \sigma_l \circ W_l f(\mathbf{x}_i) \right\|$$

$$= \frac{1}{m} \mathbb{E} \sup_{f \in \mathcal{F}_{l-1}} \sup_{W_l} \sqrt{\sum_{j=1}^n \left( \sum_{i=1}^m \epsilon_i \sigma_l \left( \mathbf{w}_j^\top f(\mathbf{x}_i) \right) \right)^2},$$

where $\mathbf{w}_j$ is the $j$-th row of $W_l$. Each matrix in the set $\{W : ||W||_F \leq B_l\}$ is composed of rows, whose sum of squared norms is at most $B_l^2$. Thus, the set can be equivalently defined as the set of

matrices, where each row $j$ equals $v_j \mathbf{w}_j$ for some $v_j > 0, ||\mathbf{w}_j|| \leq 1$, and $||\mathbf{v}||^2 \leq B_l^2$. Noting that each $v_j$ is positive, we can upper bound the expression in the displayed equation above as follows:

$$\frac{1}{m} \mathbb{E} \sup_{f \in \mathcal{F}_{l-1}} \sup_{\mathbf{w}_j, v} \sqrt{\sum_{j=1}^{n} \left( \sum_{i=1}^{m} \epsilon_i \sigma_l \left( v_j \mathbf{w}_j^\top f(\mathbf{x}_i) \right) \right)^2}$$

$$= \frac{1}{m} \mathbb{E} \sup_{f \in \mathcal{F}_{l-1}} \sup_{\mathbf{w}_j, v} \sqrt{\sum_{j=1}^{n} v_j^2 \left( \sum_{i=1}^{m} \frac{\epsilon_i}{v_j} \sigma_l \left( v_j \mathbf{w}_j^\top f(\mathbf{x}_i) \right) \right)^2}$$

$$\leq \frac{1}{m} \mathbb{E} \sup_{f \in \mathcal{F}_{l-1}} \sup_{\mathbf{w}_j, v, v'} \sqrt{\sum_{j=1}^{n} {v_j'}^2 \left( \sum_{i=1}^{m} \frac{\epsilon_i}{v_j} \sigma_l \left( v_j \mathbf{w}_j^\top f(\mathbf{x}_i) \right) \right)^2},$$

Where $||\mathbf{v}'||^2 \leq B_l^2$ (note that $\mathbf{v}$ must also satisfy this constraint). Moreover, for any choice of $\epsilon, \mathbf{v}, f$ and $\mathbf{w}_1, ..., \mathbf{w}_n$, the supremum over $\mathbf{v}'$ is clearly attained by letting $v'_{j^*} = B_l$ for some $j^*$. Plugging this observation back, we can upper-bound the displayed equation by

$$\frac{B_l}{m} \mathbb{E} \sup_{f \in \mathcal{F}_{l-1}} \sup_{\mathbf{w}_j, \mathbf{v}} \max_{j} \left| \sum_{i=1}^{m} \frac{\epsilon_i}{v_j} \sigma_l \left( v_j \mathbf{w}_j^\top f(\mathbf{x}_i) \right) \right|$$

$$= \frac{B_l}{m} \mathbb{E} \sup_{f \in \mathcal{F}_{l-1}} \sup_{\mathbf{w}: ||\mathbf{w}|| \leq 1, v \in (0, B]} \left| \sum_{i=1}^{m} \frac{\epsilon_i}{v} \sigma_l \left( v \mathbf{w}^\top f(\mathbf{x}_i) \right) \right|$$

$$= \frac{B_l}{m} \mathbb{E} \sup_{f \in \mathcal{F}_{l-1}} \sup_{\mathbf{w}: ||\mathbf{w}|| \leq 1, v \in (0, B]} \left| \sum_{i=1}^{m} \epsilon_i \psi_v \left( \mathbf{w}^\top f(\mathbf{x}_i) \right) \right|, \tag{11}$$

where $\psi_v(z) = \frac{\sigma_l(vz)}{v}$ for any $z \in \mathbb{R}$. Since $\sigma_l$ is L-Lipschitz, it follows that $\psi_v(\cdot)$ is also L-Lipschitz regardless of $v$, since for any $z, z' \in \mathbb{R}$,

$$|\psi_v(z) - \psi_v(z')| = \frac{|\sigma(vz) - \sigma(vz')|}{v} \leq \frac{L|vz - v'z|}{v} = L|z - z'|.$$

As a result, we can upper-bound Eq. 11 by

$$\frac{B_l}{m} \mathbb{E} \sup_{f \in \mathcal{F}_{l-1}} \sup_{\mathbf{w}: ||\mathbf{w}|| \leq 1, \psi \in \Psi_L} \left| \sum_{i=1}^{m} \epsilon_i \psi \left( \mathbf{w}^\top f(\mathbf{x}_i) \right) \right|,$$

where $\Psi_L$ is the class of all L-Lipschitz functions which equal 0 at the origin. To continue, it will be convenient to get rid of the absolute value in the displayed expression above. This can be done by noting that the expression equals

$$= \frac{B_l}{m} \mathbb{E} \sup_{f \in \mathcal{F}_{l-1}} \sup_{\mathbf{w}: ||\mathbf{w}|| \leq 1, \psi \in \Psi_L} \max \left\{ \sum_{i=1}^{m} \epsilon_i \psi \left( \mathbf{w}^\top f(\mathbf{x}_i) \right), - \sum_{i=1}^{m} \epsilon_i \psi \left( \mathbf{w}^\top f(\mathbf{x}_i) \right) \right\}$$

$$\overset{(*)}{\leq} \frac{B_l}{m} \mathbb{E} \sup_{f \in \mathcal{F}_{l-1}} \left[ \sup_{\mathbf{w}: ||\mathbf{w}|| \leq 1, \psi \in \Psi_L} \sum_{i=1}^{m} \epsilon_i \psi \left( \mathbf{w}^\top f(\mathbf{x}_i) \right) + \sup_{\mathbf{w}: ||\mathbf{w}|| \leq 1, \psi \in \Psi_L} - \sum_{i=1}^{m} \epsilon_i \psi \left( \mathbf{w}^\top f(\mathbf{x}_i) \right) \right]$$

$$\overset{(**)}{=} \frac{2B_l}{m} \mathbb{E} \sup_{f \in \mathcal{F}_{l-1}} \sup_{\mathbf{w}: ||\mathbf{w}|| \leq 1, \psi \in \Psi_L} \sum_{i=1}^{m} \epsilon_i \psi \left( \mathbf{w}^\top f(\mathbf{x}_i) \right),$$

where $(*)$ follows from the fact that $max\{a, b\} \leq a + b$ for non-negative $a, b$ and the observation that the supremum is always non-negative (it is only larger, say, than the specific choice of $\psi$ being the zero function), and $(**)$ is by symmetry of the function class $\Psi_L$ (if $\psi \in \Psi_L$, then $-\psi \in \Psi_L$ as well).

Note that for every $f \in \mathcal{F}_{l-1}$ and $w$ with $||\mathbf{w}|| \leq 1$ we have $|\mathbf{w}^\top f(\mathbf{x}_i)| \leq ||\mathbf{w}^\top|| \cdot ||f(\mathbf{x}_i)|| \leq R_{l-1}$, where $R_{l-1} = b_x L^{l-1} ||W_{l-1}|| \cdot ||W_{l-2}|| \cdot \cdots ||W_1||$. This class is a subset of the class of composition of all functions from $\mathbb{R}^d$ to $[-R_{l-1}, R_{l-1}]$, and all univariate $L$-Lipschitz functions crossing the

origin. Fortunately, the Rademacher complexity of such composed classes was analyzed in Golowich et al. [2018] for a different purpose. Applying Theorem 4 from that paper, we get the upper bound of

$$2cB_l L \left( \frac{R_{l-1}}{\sqrt{m}} + \log^{\frac{3}{2}}(m) R_m(\mathbf{w}^\top F_{l-1}) \right)$$

where $c > 0$ is a universal constant, and

$$R_m(\mathbf{w}^\top F_{l-1}) := \mathbb{E} \sup_{w : \|\mathbf{w}\| \le 1, f \in F_{l-1}} \frac{1}{m} \sum_{i=1}^{m} \epsilon_i \mathbf{w}^\top f(\mathbf{x}_i) \le \mathbb{E} \sup_{f \in F_{l-1}} \left\| \frac{1}{m} \sum_{i=1}^{m} \epsilon_i f(\mathbf{x}_i) \right\|$$

From this, the result follows. $\qquad \square$

*Proof of Theorem 7.* Remember the new definition of $\mathcal{F}_k$ (see Eq. 10), note that we just removed $\mathbf{w}_k$, and therefore we turn to analyze the Rademacher complexity of

$$\{\mathbf{w}_k^\top f : \|\mathbf{w}_k\| \le B_k, f \in \mathcal{F}_{k-1}\}$$

on $m$ inputs from $\{\mathbf{x} \in \mathbb{R}^d : \|\mathbf{x}\| \le b_x\}$. Fix some set of inputs $\mathbf{x}_1, \dots \mathbf{x}_m$ with norm at most $b_x$. The Rademacher complexity equals

$$\mathbb{E} \sup_{f \in \mathcal{F}_{k-1}} \sup_{\mathbf{w}_k} \frac{1}{m} \sum_{i=1}^{m} \epsilon_i \mathbf{w}_k^\top f(\mathbf{x}_i) \le B_k \cdot \mathbb{E} \sup_{f \in \mathcal{F}_{k-1}} \left\| \frac{1}{m} \sum_{i=1}^{m} \epsilon_i f(\mathbf{x}_i) \right\|. \qquad (12)$$

By applying Lemma 11 repeatedly (i.e. $k-1$ times) we get

$$\mathbb{E} \sup_{f \in F_{k-1}} \left\| \frac{1}{m} \sum_{i=1}^{m} \epsilon_i f(\mathbf{x}_i) \right\|$$

$$\le 2cB_{k-1}L \left( \frac{R_{k-2}}{\sqrt{m}} + \log^{\frac{3}{2}}(m) \mathbb{E} \sup_{f \in F_{k-2}} \left\| \frac{1}{m} \sum_{i=1}^{m} \epsilon_i f(\mathbf{x}_i) \right\| \right)$$

$$\le \frac{2cLB_{k-1}R_{k-2}}{\sqrt{m}} + \frac{(2cL)^2 B_{k-1}B_{k-2}R_{k-3}\log^{\frac{3}{2}}(m)}{\sqrt{m}} + \cdots$$

$$\cdots + \frac{(2cL)^{k-1} \left( \prod_{j \le k-1} B_j \right) \log^{\frac{3(k-2)}{2}}(m)}{\sqrt{m}} + (2cL)^{k-1} \left( \prod_{j \le k-1} B_j \right) \log^{\frac{3(k-1)}{2}}(m) \mathbb{E} \left\| \frac{1}{m} \sum_{i=1}^{m} \epsilon_i \mathbf{x}_i \right\|,$$
$$(13)$$

where $R_{k-2} = b_x L^{k-2} \|W_{k-2}\| \cdot \|W_{k-3}\| \cdot \cdots \cdot \|W_1\|$ and $R_0 = b_x$. Note that by Cauchy-Schwarz inequality

$$\frac{1}{m} \mathbb{E} \left\| \sum_{i=1}^{m} \epsilon_i \mathbf{x}_i \right\| \le \frac{1}{m} \sqrt{\mathbb{E} \left\| \sum_{i=1}^{m} \epsilon_i \mathbf{x}_i \right\|^2} = \frac{1}{m} \sqrt{\mathbb{E} \sum_{i,i'=1}^{m} \epsilon_i \epsilon_{i'} \mathbf{x}_i^\top \mathbf{x}_{i'}} = \frac{1}{m} \sqrt{\sum_{i=1}^{m} \|\mathbf{x}_i\|^2} \le \frac{1}{m} \sqrt{m b_x^2} = \frac{b_x}{\sqrt{m}}.$$

Plugging this back into Eq. 13, we get the following upper bound:

$$\frac{2cLB_{k-1}R_{k-2}}{\sqrt{m}} + \frac{(2cL)^2 B_{k-1}B_{k-2}R_{k-3}\log^{\frac{3}{2}}(m)}{\sqrt{m}} + \cdots +$$

$$\cdots \frac{(2cL)^{k-1} \prod_{j \le k-1} B_j \log^{\frac{3(k-2)}{2}}(m)}{\sqrt{m}} + \frac{(2cL)^{k-1} \prod_{j \le k-1} B_j \log^{\frac{3(k-1)}{2}}(m) b_x}{\sqrt{m}}$$

$$= \sum_{i=1}^{k-1} \frac{(2cL)^i \cdot \left( \prod_{j=k-i}^{k-1} B_j \right) R_{k-1-i} \cdot \log^{\frac{3(i-1)}{2}}(m)}{\sqrt{m}} + (2cL)^{k-1} \prod_{j \le k-1} B_j \log^{\frac{3(k-1)}{2}}(m) \frac{b_x}{\sqrt{m}}.$$

Altogether we have $k$ terms, each one of them upper bound by

$$\frac{(2cL)^{k-1} R_{k-2} \left( \prod_{i \le k-1} B_i \right) \log^{\frac{3(k-1)}{2}}(m)}{\sqrt{m}},$$

where we use the assumption that $L$ and $||W_i|| \geq 1$, which implies also that $||W_i||_F \geq 1$. Therefore, we can upper bound the displayed Eq. by

$$\frac{k(2cL)^{k-1}R_{k-2}\left(\prod_{i\leq k-1}B_i\right)\log^{\frac{3(k-1)}{2}}(m)}{\sqrt{m}}.$$

Combining with Eq. 12, the Rademacher complexity of $\mathcal{F}_k$ is upper bounded by

$$\frac{k(2cL)^{k-1}bR_{k-2}\left(\prod_{i\leq k-1}B_i\right)\log^{\frac{3(k-1)}{2}}(m)}{\sqrt{m}},$$

from which the result follows. $\qquad\square$

