# OpenReview forum: "Initialization-Dependent Sample Complexity of Linear Predictors and Neural Networks"
_NeurIPS.cc/2023/Conference — NeurIPS 2023 poster_

### Official Review · Reviewer_xxjV · 2023-06-27

**Soundness:** 4 excellent
**Presentation:** 2 fair
**Contribution:** 3 good
**Rating:** 7
**Confidence:** 3

**Summary:**

This paper views a prototype but interesting enough neural network architecture from Rademacher complexity, and uses Rademacher complexity to analyze the sample complexity used for correctly training the neural network. It reveals an approves an interesting result that the Rademacher complexity hence the sample complexity depends on the initialization. The major theorems show the difference of sample complexity bound at zero and non-zero initialization.

=============

Rating updated to 7 and Confidence to 3 for the rebuttal. I would be glad to see it published, but the tightness given by shattering, as pointed out by the authors, relies on the "distribution-free learning setting", which might not be practical when considering the real world data (whose distribution might make some neural net architectures more learnable), and does not talk about the optimization algorithm that trains the model. The manuscript is still self-consistent so I give 7, but not higher due to the unanswered larger scope.

If the real world experiment shows the gap and difference between two initializations, I would not hesitate to give 7 or 7+, but now I'm still hesitant between 6 and 7.

**Strengths:**

I think the mathematics is concrete and solid, the contribution is novel, and it shows an interesting and non-intuitive fact that sample complexity depends on the initialization. All the mathematical definition are rigorously and clearly defined, and there is thorough discussion of the impact of the results. The special bound for deep neural networks is also given as a corollary.

**Weaknesses:**

I would like to see more practical experiments that verify the consistency between sample complexity and the Rademacher complexity. Rademacher complexity, especially for deep neural networks or functions with high nonlinearity, is usually not tight for sample complexity bound. Rademacher complexity is an upper bound, i.e., if you have more samples and can find the globally optimal model in the model family, then it’s guaranteed that the model is the ground truth. But it does not say 1) what if number of samples is lower than sample complexity bound, how does the generalization error grow, which depends on the data distribution; 2) if there are enough samples, is the neural net trained with typical methods like SGD the global optimal solution. Those need to be verified by experiments, no need to be exactly consistent, but it cannot be a too large gap.

**Questions:**

I did not understand the difference between the results in Section 3.1 and 3.2, they look quite similar. I would like to see more discussion, corollaries or remarks following the theorems, since they are not described as "sample complexity", but "shatter", etc.

I would be willing to increase the score if the weaknesses can be addressed.

---

> ### Author Rebuttal · Authors · 2023-08-03
>
> Thanks for your comments. We will see whether we can incorporate experiments (say along the lines of Bartlett et al [2017]). However, it should be emphasized that our paper is theoretical and focuses on understanding the minimax optimal sample complexity of various predictor classes, similar to many previous papers in the statistical learning theory literature.
> - Difference between sections 3.1, 3.2: The main difference is that 3.1 focuses on the case of zero initialization ($W_0=0$) and 3.2 focuses on the case of non-zero initialization ($W_0\neq 0$). These settings are qualitatively different, as we show that the sample complexity can be finite in the former case, but infinite in the latter case.

---

> > ### Comment · Reviewer_xxjV · 2023-08-12
> > **Thanks for the comment, and follow-up**
> >
> > I do agree with the authors' point, but my question is not about just verifying by and experiment, but asking whether the Rademacher complexity is a tight bound -- if the authors can theoretically  prove it, say, giving a made-up example and prove it achieves the bound that is exactly the same as the theory, then it would even be stronger than an experiment.
> >
> > Maybe I'm not familiar with the convention in this area, lift score to 5 in case tightness is typically not a concern.
> >
> > Yes, 3.1 and 3.2 are about different initialization assumptions, but I'm a bit confused by the proposition statement. I guess one is a positive result and another is the opposite, could the authors explain more about the different in conclusion (rather than assumption)?

---

> > > ### Author Response · Authors · 2023-08-13
> > >
> > > - Indeed the Rademacher complexity just upper bounds the sample complexity. Therefore Thm. 2 just states that the sample complexity is at most $2^{O(B^2/\epsilon^2)}$, independent of the size/dimension parameters $n$ and $d$. This is why we also prove a tight lower bound of $2^{\Omega(B^2/\epsilon^2)}$, by lower bound the fat-shattering dimension (i.e. Thm. 1), which characterized the sample complexity. To see an example where the sample complexity is the same as the theory, we refer to the proof of the theorem that gives a lower bound on the sample complexity in terms of the fat-shattering dimension, which is analogous to the proof of the VC dimension in a classification task and can be found for example in Anthony and Bartlett [2002]: Page 262, part 3, subsection 19.5, Thm. 19.5. See line 123 in our paper for the full reference.
> > >
> > > - In section 3.1 with zero initialization, we show that the sample complexity is $2^{\Theta(B^2/\epsilon^2)}$, independent of the size/dimension parameters $n$ and $d$. In contrast, in the case of non-zero initialization in section 3.2, we show that the sample complexity is infinite in a size-independent setting i.e. the sample complexity depends on the size of the model $n$ and $d$, even if the initialization is very small. This is perhaps a surprising result, since in other models, like the class of scalar-valued linear predictors composed with some Lipschitz function, the initialization doesn't affect the sample complexity.

---

### Official Review · Reviewer_Zmxy · 2023-07-06

**Soundness:** 3 good
**Presentation:** 2 fair
**Contribution:** 3 good
**Rating:** 6
**Confidence:** 3

**Summary:**

In this paper, the authors give sample complexity bounds for learning function classes of the form $g(x) = f(Wx)$, where $W$ is a matrix, and $f$ is a Lipshitz function. They sample complexity bounds are based on the Frobeneous norm of the matrix $W - W_0$, where $W_0$ is a fixed initialization matrix. The upper bounds are given in terms of the Rademacher complexity, which the lower bounds are given in terms of the fat-shattering dimension. For the case when $W_0 = 0$, the authors give tight, size independent bounds on the sample complexity, where here size refers to the first dimension, $n$, of $W$. They use this to show some novel sample complexity bounds for neural networks. The then show that when $W_0 \neq 0$, it is impossible to achieve such size-independent bounds, and the resulting sample complexity that grows in $m$. Due to this impossibility result, they are able to show the existence of a certain type of ERM problem for which SGD learns but uniform convergence fails.  Finally they restrict their setting to the case of two-layer neural networks and show some specific results that recover or improve upon results in the literature.

**Strengths:**

In general the paper is reasonably clear, and the results are solid. They authors close some open questions in related work, and provide novel results in more general settings.

**Weaknesses:**

The results in this paper are somewhat incremental in the sense that they seem to primarily just be useful for the case of a one-layer network with multiple outputs (and practically, in such a setting, the "size" parameter $n$ would probably not be so large). Could the authors make the exact dependence on $n$ clear in the resulting sample complexity? (I think it is logarithmic?)

*Lacking Motivation*:
- The model class $f(Wx)$ studied in this work seems to be different from previous classes studied from a sample complexity perspective, and so the authors should further justify their setting, and why it differs from existing models. Are there settings that generalize this model? And which other settings does this model subsume besides the vector-valued case? The examples given in the paragraph at line 13 could be made even more concrete and obvious to the author.
- The authors should provide some references in the intro for where such settings are studied in other works or where they are used in practice.

*Comparison with related work could be improved*:
- Intro: The authors should explain the result of Vardi et al 22 and Daniely and Granot in the introduction, and specifically explain how their work differs, because it is quite similar. It seems like the major difference is that f has a decomposable structure in the other works? The authors should also spell out the exact open questions they answer from Vardi et al.
- The authors should include a detailed quantitative comparison to the most related works. A table (perhaps in the intro, though if it is too complicated, later on) would be helpful, describing the different settings and comparing results ($W_0 = 0$ or not, shallow or deep nets, element-wise activations, etc.).
- For the result on deep neural networks on page 6 and in section 4.1, a more detailed quantitative comparison to existing works would be useful. In what regimes are the bounds given in this work better? In what regimes are other works in the literature stronger?
- More discussion on the limitations of uniform convergence (UC) and works that go beyond UC would be useful. Eg. https://arxiv.org/abs/1902.04742 , https://arxiv.org/abs/2103.04554, https://arxiv.org/abs/2206.07892 and references therein. The authors should discuss other way of bounding the sample complexity of learning neural networks eg. PAC Bayes bounds etc.


**Questions:**

Major comments are in weaknesses.
Minor comments:
- In line 68, could the authors explain what they mean by the Lipchitz function is a parameter? If possible, this should be explained concisely, but otherwise a pointer to where it is explained later would be helpful.
- It is standard to have $n$ be the number of samples and $m$ be a width parameter, so it would clearer if these were swapped

**Limitations:**

Sufficient

---

> ### Author Rebuttal · Authors · 2023-08-03
>
> Thanks for your comments, we will improve the presentation according to your suggestions.
> - Show dependence on $n$: We would like to emphasize that our focus is on size-independent bounds, which do not depend in any manner whatsoever on $n$. This is in line with a huge previous literature in statistical learning theory, as well as on neural networks in particular. Such results are relevant for understanding the generalization capabilities of large models, and are also crucial for understanding how different norms lead to sample complexity control, independent of additional orthogonal constraints such as model size. Understanding how the bounds are affected by joint norm control and model size control is certainly an interesting avenue for future research, but we believe a necessary prerequisite is understanding the effect of each separately.
> - Motivation of model class: We will add further discussion. In a nutshell, it directly models vector-valued linear predictors composed with a Lipschitz loss (e.g., for multi-task and multi-class learning), as well as generalizes large neural networks. The former is of course studied in many papers. As to the latter, an example reference is Daniely and Granot [2022]. In any case, this model helps us to understand the effect of initialization on the sample complexity (which is a challenging task compared to the case of initialization at zero), the sample complexity of neural networks, and the conditions under which we can achieve size-independent bounds.
> - Comparison to previous work: We will add more discussions as suggested, and also include a summary below. The open questions in Vardi et al that we answer are (1)  Understanding the sample complexity with non-zero initialization (e.g., Thm. 6, see lines 336-337, 352-354), and (2) Extending their results to deeper networks (e.g., Thm. 7, see lines 368-370,378-389). As to Daniely and Granot, their bounds are not size-independent, so it is a different setting than ours.
> - Line 68: We meant that the predictor class ranges over all possible $L$-Lipschitz functions, composed with a class of linear functions (as opposed to having a single fixed Lipschitz function). See exact definition in the statement of Theorem 2.
>
>
>
>
> Further discussion of related works:
>
> - Deep neural networks and Frobenius norm: We refer to lines 333-342.
> In more detail: A width-independent uniform convergence
> guarantee, depending on the product of Frobenius norm of all layers, has been established in Neyshabur et al. [2015] for
> constant-depth networks, and in Golowich et al. [2018] for arbitrary-depth networks. However, these bounds are specific to element-wise, homogeneous activation functions, whereas we tackle general Lipschitz activations. Bounds based on other norms include Anthony
> and Bartlett [1999], Bartlett et al. [2017], Liang [2016], but are potentially more restrictive than the
> Frobenius norm, or do not lead to width-independence. As we mention on page 6 (lines 226-230), all previous bounds of this type we are aware of strongly depend on various norms of all layers, which can be arbitrarily larger than the spectral norm in a size-independent setting (such as the Frobenius norm and the $(1,2)-$norm), or made strong assumptions on the activation function.
>
> - Non-zero initialization: Bartlett et al. [2017] upper bound the sample complexity of neural networks with non-zero initialization, but they used a much stronger assumption than ours: They control the $(1,2)$-matrix norm, whereas the gap between this norms can be arbitrarily large, depending on the matrix size. Vardi et al [2022] also studied the initialization case with element-wise activations, but their result is size-dependent and includes a different technique than ours.
>
> - Non-element-wise activations: Daniely and Granot [2022] do provide a fat-shattering lower bound with a general Lipchitz activation (non-element-wise, similar setting as our first part of the paper), which implies that neural networks on $R^d$ with bounded Frobenius norm and width $n$ can shatter $n$ points with constant margin, assuming that the inputs have norm at most $\sqrt{d}$ and that $n = O(2^d)$.
> However, this lower bound is size-dependent, and moreover does not separate between the input norm bound and the width of the hidden layer. Therefore, their result does not contradict our upper bound (i.e. Thm. 2) which says that it's possible to achieve a size-independent upper bound on the sample complexity.

---

> > ### Comment · Reviewer_Zmxy · 2023-08-19
> >
> > I have read the response of the authors; thank you for the discussion of related work.
> >
> > For my question about size-dependence, I meant for the case when $W_0 \neq 0$, though I suppose there is no matching upper bound on sample complexity in that setting? Do the authors expect there to be a sample complexity bound that matches Theorem 3?

---

> > > ### Author Response · Authors · 2023-08-19
> > >
> > > In the case of $W_0 \neq 0$, we don't have a tight upper bound in terms of n and d. We emphasize the message of Thm. 3 that it is impossible to control the sample complexity independent of the size/dimension parameters n and d. We believe that
> > > using standard covering number arguments can achieve an upper bound that is polynomial on $n$. It remains to be seen whether the actual dependence on $n$ is polynomial, logarithmic, or something else. In any case, this is an interesting question and we will add it to the open questions (i.e. to subsection 5). Thanks.

---

### Official Review · Reviewer_8WAD · 2023-07-07

**Soundness:** 3 good
**Presentation:** 4 excellent
**Contribution:** 3 good
**Rating:** 7
**Confidence:** 2

**Summary:**

The paper generally studies the sample complexity of the functions of the form $f(Wx)$ and it particularly focuses on size-independent bounds. First it shows matching exponential lower and upper bounds for the case that the reference matrix $W_0=0$. Then, it is showed that one cannot obtain an upper bound in the case that $W_0 \neq 0$. Using these results, they further provide: (1) sample complexity bounds for NNs and (2) an instance where uniform convergence does not hold but learning is possible with SGD.

**Strengths:**

- The paper generally studies the sample complexity of $f(Wx)$ quite extensively both in the case that the reference matrix $W_0$ is zero and non-zero. The implications are also interesting:
    - Neural network results: e.g., depth/width-independent result if the reference matrix is 0
    - Another example that uniform convergence fails and SGD succeeds.
- Both the main text and the appendix are well-written.

**Weaknesses:**

My main concern is regarding Q1 below. There are some limitation/future directions that have been discussed in the paper itself.

**Questions:**

Main question:
- Q1. In theorem 1, the valued of $d, n$ are asymptomatically determined by $L^2B^2 /  \epsilon^2$ which is the exact quantity that gives the shattering. In other words shattering in this case is given by $d$ (or $n$) as well. So I am not sure if this lower bound is independent of the size? (Both the bound and the size depend on $L^2B^2 /  \epsilon^2$).

Other questions:
- Q2. (Although this is not super important for the negative result) Do we have a picture about Theorem 3, 5 when $n$ is not exponentially large w.r.t. $d$?

Possible typos:
- line 232: $W_j$ instead of $w_j$
- line 234: do not instead of don't
- line 422: $x_1, \ldots, x^m$
- line 633: space between $R$ and (


**Limitations:**

The work is theoretical and does not have any negative societal impact. The limitations have been adequately discussed.

---

> ### Author Rebuttal · Authors · 2023-08-03
>
> Thanks for your comments, we will fix the typos.
>
> - Q1: To prove a lower bound in a size-independent setting, we are free to choose the size parameters $n,d$ as we wish (since the upper bounds should hold for any $n,d$). In particular, we may choose them to depend on $L,B,\epsilon$. It would be interesting to understand the sample complexity when $n,d$ are also controlled, and we left this as an open question (see section 5).
> - Q2: That's an interesting question for future research. In that case we would attain some dependence on $n$ (but whether it is polynomial, logarithmic or something else remains to be seen).

---

> > ### Comment · Reviewer_8WAD · 2023-08-16
> >
> > Thank you for your response. Could you please clarify more on Q1? Particularly, in Theorem 1, shattering is $\exp(\Theta(d))$ or $\mathrm{poly}(n)$, so the bound depends on dimension/size? (This is related to the first weakness raised by Reviewer 9YyA.)

---

> > > ### Author Response · Authors · 2023-08-16
> > >
> > > In Thm. 1 we showed that for $d = \Theta(B^2/\epsilon^2), n = exp (\Theta(d))=exp(\Theta(B^2/\epsilon^2))$ our class can shatter $exp(\Theta(B^2/\epsilon^2))$ points. It doesn't mean that our bound depends on dimension/size, since $n$ and $d$ also depend on the norm $B$. Without this dependence, the result no longer holds.
> > > You may be wondering for example if we can use the same technique to choose any $n$ (say $n >> exp(\Theta(B^2/\epsilon^2))$), and show that our class can shatter $n$ points, which means that the sample complexity also depends on the dimension/size and lead to a contradiction to Theorem 2. The answer is negative. We refer you to the proof of Thm. 1 which is based on Lemma 2. We emphasize two things in Lemma 2: the first is that $n = exp(\Theta(d))$ and the second is that $||W_s||_F^2 \leq 2d$. This establishes the dependence between the norm $B$, to $n$ and $d$. If necessary, we can also explain why Lemma 2 no longer holds if we remove the assumption that $n = exp(O(d))$.

---

> > > > ### Comment · Reviewer_8WAD · 2023-08-20
> > > >
> > > > Thank you for the additional explanations. I understand better now and I will keep my score.

---

### Official Review · Reviewer_9YyA · 2023-07-11

**Soundness:** 2 fair
**Presentation:** 2 fair
**Contribution:** 2 fair
**Rating:** 3
**Confidence:** 5

**Summary:**

The authors provide various sample complexity results for linear and nonlinear networks in initialization dependent and independent cases. Here is the summary of results.
* For the class of predictors $f(Wx)$, the fat-shattering dimension is characterized and gives a lower bound on sample complexity where the dependence of the bound on the distance from initialization $B$ is exponential (Theorem 1). The output dimension, however, is assumed to be exponential in $B$: $n=\exp(\Theta(L^2B^2/\epsilon^2))$.
* The upper bound is driven by Rademacher complexity analysis in Theorem 2, and is shown to be tight.
* The proof of Theorem 2 relies on covering number arguments, Dudley’s inequality and utilizing covering bounds from Lemma 1 and 5.
* Corollary 2 extends the result of Theorem 2 to neural networks by replacing the function $f$ by all the layers and activations after the first layer. The Lipschitz constant of the new function is equal to the product of spectral norms.
* Theorem 3 shows that the fat shattering lower bound holds as well for non-zero initialization, although $\Vert W_0\Vert$ should be as small as $2\sqrt{2}\epsilon$.
* Theorem 4 adapts the classical result for convex learning in context of learning vector-valued linear predictors. Theorem 5 adapts Theorem 3 for convex functions. The overall message is that linear predictors combined with convex functions are learnable.
* The authors provide generalization bounds to single hidden layer networks with elementwise Lipschitz activation function and with initialization dependence in Theorem 6. This is the extension and improvement of Vardi et al 2022, and Daniely and Garnot 2022.
* Theorem 7 provides a generalization bound on deep neural networks with Lipschitz activation under Forbenius norm constraints. The proof is based on Rademacher complexity analysis, and adapts Golowich et al’s peeling lemma to Lipschitz activation.

To summarize, the paper contains various theoretical results, some related to vector valued outputs, some related to initialization dependence, some related to convex functions, and some related to Lipschitz assumption. I have some concerns about the theoretical results of the paper, their assumptions and comparison with existing works.
Below, I raise some concerns about the assumptions of Theorem 1,3 and 5. The result of Theorem 2 seems peculiar. Theorem 7 is not properly situated with respect to similar norm-based bounds in the literature. Theorem 4 and 6 seems to build heavily on previous results.


**Strengths:**


Certain story lines of the paper are very interesting, for example the impact of vector valued outputs on the generalization. Also, adapting Peeling lemma in Theorem 7 is very elegant – although it comes with an additional cost.


**Weaknesses:**

* I wonder about the message of  Theorem 1. The authors assumed already that the output dimension is exponentially large $n=\exp(\Theta(L^2B^2/\epsilon^2))$, and then show that the network can shatter $\exp(cL^2B^2/\epsilon^2)$ points. This just tells that the fat-shattering dimension would be linearly (or polynomially maybe) dependent on the output dimension and not exponentially on $B$, as it is claimed in the paper.
* It is surprising that Theorem 2 do not leave any room for reducing to scalar valued case ($n=1$), where the dependence on $B$ is not exponential anymore. Is there an explanation for this? In this sense, the results are not optimal for $n=1$. More generally, it is surprising that the dimension of the output vector $n$ does not impact the sample complexity.
* The exponential dependence arises mainly from the covering number argument in Lemma 1, where the function space is covered by binning $N_x$ and $N_y$. The exponent is basically $N_x$. It would be important to emphasize this in the main paper, explain why $B$ appears in the exponent, and whether this can be circumvented.
* I wonder if directly bound the Rademacher complexity for $f(Wx)$ is a good choice. One might be tempted to use variants of vector-valued contraction lemma  (for example Maurer’s result) to remove $f$, although with price of dimension dependence. It seems to me that this approach will not suffer from the exponential dependence.
* On a similar note, the authors should discuss the existing approaches for bounding the Rademacher complexity of vector valued functions and situate their work with respect to those.
* Corollary 2 utilizes the Lipschitz constant of the deep network to bound the generalization error. However, this has been tried already in Bartlett et al 2017 and does not have exponential dependence. Besides, considering Lipschitz constant alone without margin has shortcomings already mentioned in  Bartlett et al 2017. The proposed bound suffers from similar shortcomings apart from the exponential dependence.
* The assumptions of Theorem3 are not clearly discussed. It seems surprising that the input and output dimensions are chosen as a function of $m$ ($d=\Theta(m)$ and $n=\Theta(\exp(m))$. The norm $B$ is fixed to $\sqrt{2}$ unlike Theorem 1. Overall, it is difficult to parse the message behind Theorem 3. We can raise similar issues for Theorem 5.
* The authors claim that Theorem 7 provides “the first of this type (to the best of our knowledge) that handles general Lipschitz activations under Frobenius norm constraints”. Bartlett et al 2017 provide a generalization error for Lipschitz activations using spectral and $(p,q)$ matrix norms. Their bound has only logarithmic dependence on the width, and does not have the product of Frobenius norms, which can be much larger than the product of spectral norms. The authors should compare their result with Bartlett et al.  There are other norm based bounds in the literature, and the authors can provide a better comparison with them.
* A side note is that Theorem 1 and 2 and 7 are for $W_0=0$. If wonder the initialization dependence, which is in the title, is the core idea of the paper. The paper covers many different topics and can probably focus deeper on some of those. For example, I feel that the result about learning convex functions is expected and not very insightful.



**Questions:**

* In line 550 of supplementary materials, it is assumed that $\frac{B}{\epsilon}>1$. I guess this is because we have the condition on the rank $r\leq B^2/\epsilon^2$. How would the proof work if the condition does not hold, and then $r=0$? The point of working with the norms is that we hope $B$ is small after training.
* Line 206, “in sharp contrast to the case of vector-valued predictors” $\to$ maybe “in sharp contrast to the case of *scalar*-valued predictors”?
* Line 685  in the supplementary materials: the equality should be an inequality.


**Limitations:**

See above.

---

> ### Author Rebuttal · Authors · 2023-08-03
>
> Thanks for your careful reading of the paper and the detailed comments, we will be happy to add clarifications and discussions as suggested.
>
> First, a general comment relevant to most of the points raised: We would like to emphasize that our focus is on *size-independent* bounds, which do not depend in any manner whatsoever on the dimension or number of parameters. This is in line with a huge previous literature in statistical learning theory, as well as on neural networks in particular. Such results are relevant for understanding the generalization capabilities of large models, and are also crucial for understanding how different norms lead to sample complexity control, independent of additional orthogonal constraints such as model size. Understanding how the bounds are affected by joint norm control and model size control is certainly an interesting avenue for future research, but we believe a necessary prerequisite is understanding the effect of each separately. We believe our results certainly pass the novelty test in that regard. For example, the fact that controlling the Frobenius norm alone leads to any kind of finite sample complexity (Theorem 2) should be surprising considering what we currently know (e.g. Maurer's vector-valued contraction result, which is vacuous without controlling the vector dimension).
>
> - Message of theorem 1: Actually, the sample complexity cannot simply depend on the output dimension. This is because theorem 2 tells us that there is also an upper bound independent of the output dimension, as long as the Frobenius dimension is controlled. In any case, as written above, our focus is on bounds which do not depend on the output dimension.
> - Theorem 2 does not reduce to the scalar-valued case: Indeed it should not, since our focus is on size-independent bounds which do not depend on $n$. This is exactly analogous to the classical $\frac{B^2}{\epsilon^2}$ sample complexity bound for linear predictors, which is minimax optimal when the input dimension is unconstrained, but becomes something else if we also constrain the dimension.
> - Existing works on Rademacher complexity of vector-valued functions: We agree these should be further discussed and would be happy to add such a discussion. As far as we know, all results on this topic strongly depend on the vector dimension, and are not applicable to the size-independent setting that we focus on (we certainly tried :) ).
> - Corollary 2 and Bartlett et al. (2017): The Bartlett et al. paper uses a different and much stronger assumption that ours: They control the $(1,2)$ matrix norm, whereas we control the Frobenius norm. Besides the latter norm being arguably more natural (as we discuss in the introduction), the gap between the $(1,2)$-norm and the Frobenius norm can be arbitrarily large, depending on the matrix size (and again, we focus on size-independent bounds). In addition, we already know that the product of the spectral norms is insufficient to bound the Rademacher complexity. The main point of our corollary 2 is that *just* by adding control on the Frobenius norm of the *first* layer, we can already get some finite sample complexity. This should be surprising, as all previous norm-based bounds (that we are aware of) required stronger norm controls on *all* the layers to get a finite bound.
> - Assumptions in Theorem 3 and theorem 5: Again, our focus is on size-independent bounds, so in order to establish a lower bound, it is reasonable to choose $n$ and $d$ to get the tightest lower bound possible (including as a function of the sample size $m$). A similar assumption is used in the classical result that the $\frac{B^2}{\epsilon^2}$ bound for linear predictors is tight (i.e. the input dimension should be larger than the sample size). The message of theorem 3 is that even if the Frobenius norm $B$ is controlled (and even a small numerical constant), we cannot obtain a finite sample complexity when $W_0\neq 0$. Also, the $\sqrt{2}$ factor in theorems 3+5 is just for convenience, and can easily be replaced by $1$ via rescaling. Anyway, we would be happy to further clarify the presentation of the results if needed.
> - "The overall message [of theorems 4+5] is that linear predictors combined with convex functions are learnable." Actually, the fact that linear+convex functions are learnable is classical and appears in textbooks. The main message is that we present a class that is provably learnable, in a distribution-free learning setting, *without* uniform convergence. Namely, the convergence of the learning algorithm does not depend on the size of the model, but the sample complexity does depend.
> - Theorem 7 and comparison to Bartlett et al.: Thanks, we will add more comparisons to the literature. Again, we would like to emphasize that Barlett et al 2017 considers an incomparable setting where the $(1,2)$ matrix norm is controlled.
> - The result about learning convex functions is expected: Certainly, it is just given for completeness. The more interesting result is theorem 5 which shows that uniform convergence does not hold here.
>
> - Question 1: The main regime of interest is $\frac{B}{\epsilon}>1$, i.e. that $B$ is not too small and $\epsilon$ is not too large. If $\frac{B}{\epsilon}<1$, then $B<\epsilon$. This means that the predictor class (with a norm bound of $B$) cannot attain an output value varying by  more than $\epsilon$, even on a single fixed data point.
> - Question 2: Thanks, we will fix accordingly.
> - Question 3: Thanks, we will fix accordingly.
>
> Regarding your summary:
> - We emphasize that Thm. 3 shows that even for very small non-zero initialization, it is impossible to control the sample complexity independent of the size/dimension parameters $d,n$, by showing that the fat shattering is infinite. This is in contrast to Thm. 1 and 2 which showed a tight size-independent bound.

---

### Decision · Program_Chairs · 2023-09-21

**Decision:**

Accept (poster)

**Comment:**

This nice paper slightly continues a long line of progress on generalization bounds for deep networks. I urge the authors to use the extra camera ready page for reviewer suggestions. Minor comment: "$k$-1" before Theorem 7 should be "$(k−1)$" or "$k-1$".